# A conceptual framework for assessing socio-hydrological resilience under change

Feng Mao[1], Julian Clark[1], Timothy Karpouzoglou[2], Art Dewulf[2], Wouter Buytaert[3, 4], and David Hannah[1]

[1] School of Geography, Earth and Environmental Sciences, University of Birmingham, Birmingham, UK

[2] Public Administration and Policy Group, Wageningen University, Wageningen, Netherlands

[3] Department of Civil and Environmental Engineering, Imperial College London, London, UK

[4] Grantham Institute for Climate Change and the Environment, Imperial College London, London, UK

*Correspondence to*: Feng Mao (f.mao@bham.ac.uk)

# Abstract

Despite growing interest in resilience, there is still significant scope for increasing its conceptual clarity and practical relevance in socio-hydrological contexts. Specifically, questions of how socio-hydrological systems respond to and cope with perturbations and how these connect to resilience remain unanswered. In this opinion paper, we propose a novel conceptual framework for understanding and assessing resilience in coupled socio-hydrological contexts, and encourage debate on the inter-connections between socio-hydrology and resilience. Taking a systems perspective, we argue resilience is a set of systematic properties with three dimensions: *absorptive, adaptive* and *transformative*, and contend that socio-hydrological systems can be viewed as various forms of human-water couplings, reflecting different aspects of these interactions. We propose a framework consisting of two parts. The first part addresses the identity of socio-hydrological resilience, answering questions such as 'resilience of what in relation to what'. We identify three existing framings of resilience for different types of human-water systems and subsystems, which have been used in different fields: (1) the *water subsystem,* highlighting *hydrological resilience* to anthropogenic hazards; (2) the *human subsystem,* foregrounding *social resilience* to hydrological hazards; and (3) the *coupled human-water system,* exhibiting *socio-hydrological resilience*. We argue that these three system types and resiliences afford new insights into the clarification and evaluation of different water management challenges. The first two types address hydrological and social states, while the third type emphasises the feedbacks and interactions between human and water components within complex systems subject to internal or external disturbances. In the second part, we focus on resilience management and develop the notion of the 'resilience canvas', a novel heuristic device to identify possible pathways and to facilitate the design of bespoke strategies for enhancing resilience in the socio-hydrological context. The 'resilience canvas' is constructed by combining absorptive and adaptive capacities as two axes. At the corners of the resulting two-dimensional space are four quadrants which we conceptualise as representing resilient, vulnerable, susceptible, and resistant system states. To address projected change-induced uncertainties, we recommend effort is now focused on shifting socio-hydrological systems from resistant towards resilient status. In sum, the novel framework proposed here clarifies the ambiguity inherent in socio-hydrological resilience, and provides a viable basis for further theoretical and practical development.

*Keywords*: *water, adaptive management, socio-hydrological system, pathway, resilience*

# 1 Introduction

There is now great interest in understanding human-water relationships from a complexity perspective. One example is the field of 'hydro-sociology' (Linton and Budds, 2014; Sivakumar, 2012), emphasising social scientific and humanities approaches to understanding the interactions between humans and water. Similarly, Sivapalan *et al*. (2012, 2014) have foregrounded the human role in the water cycle by establishing 'socio-hydrology' as a perspective to understand modification and changing patterns of water use in the Anthropocene. While presenting hydrological complexity from different viewpoints, both approaches highlight the interrelationship of human and water systems as one prone to instability. Shifting hazard regimes and altering external conditions caused by human-induced change means dealing with uncertainties, and the prospect of system degradation to undesired states and/or collapse. This opens up questions of how socio-hydrological systems respond to perturbations and future management uncertainties, making it opportune to explore the concept of resilience in socio-hydrological contexts.

Since its introduction in the 1970s, the idea of resilience has evolved from a descriptive notion to a normative concept with broad and often ambiguous meanings (Brand and Jax, 2007; Olsson et al., 2015; Weichselgartner and Kelman, 2015). Some authors have observed this lack of conceptual clarity hinders the application of resilience thinking (Strunz, 2012), raising questions of how to apply the concept to socio-hydrological systems. In the coupled human-water context, resilience is now used in many different ways, such as hydrological resilience, aquatic ecological resilience, community and urban resilience to hydrological disasters, and resilience of water cycles (Rockström et al., 2014). Yet these applications do not always capture the essence of socio-hydrological dynamics or lend support to inter-disciplinary resilience research. We argue that this is because of our limited understanding of human-water couplings and hence the type of resilience that adheres to systems, as much as it is a product of lack of clarity in analysing systematic change. As a result, system identities need defining before examination is undertaken of their intrinsic resilience types (Cumming et al., 2005).

Our aim here is to propose a conceptual framework for assessing resilience in socio-hydrological contexts, and by which we provide opinions for understanding and managing socio-hydrological resilience. Instead of offering a single prescriptive solution, this framework supports pluralist perspectives and encourages debate on socio-hydrology and its interrelations with resilience. The paper's structure is as follows. In Section 2, we examine the relationship between resilience, system type and capacities, and characterise resilience as a set of *absorptive, adaptive* and *transformative* properties of the system. In Section 3, we classify three types of human-water couplings and their associated resilience forms. Within this classification, we propose studying socio-hydrological resilience, and explain how this differs from the existing notions of hydrological and social resilience. We also show how different resilience capacities arise in each human-water coupling type. We proceed to argue socio-hydrological systems and subsystems and their attendant resilience dynamics can be characterised using the conceptual toolkit of ecosystem services, as this approach effectively classifies dominant processes of human-water interactions already. In Section 4, we show how to implement the new concept of socio-hydrological resilience. To do so, we develop the notion of a 'resilience canvas' to specify pathways toward particular socio-hydrological resilience states. We analyse the role of different resilience capacities in water management under change, and identify fields of further enquiry. Our overall conclusion is that to enhance system capacity to face future uncertainties requires a concerted shift to move from resistance to resilient water management.

# 2 Resilience, systems and capacities

The concept of resilience has many definitions, and is routinely used in multiple fields in widely contrasting contexts (Brand and Jax, 2007). Our aim here is not to focus on this variety, but instead to characterise how resilience is interpreted in order to clarify its relationship to other concepts especially systemic capacities and properties (Anderies et al., 2004; Klein et al., 2003; Plummer and Armitage, 2007). Resilience is usually defined as the capacity of a system to absorb disturbance without

substantially challenging its function or structure (Walker et al., 2004). In a more generalised definition, resilience is 'the capacity to persist in the face of change, to continue to develop with ever changing environments' (Folke, 2016 p.2). Thus, this concept is understood as a set of systemic *absorptive, adaptive* and *transformative* capacities, which offers scope for its conceptualisation in three dimensions – persistence for now, and response for future contingencies in incremental or in radical ways (Béné et al., 2014; Miller et al., 2011). A clear understanding of the overall *system* is thus an essential precursor to any evaluation of its resilience, since it determines what the identity or subject of resilience is (Carpenter et al., 2001; Evans and Reid, 2013). A system refers to a set of interacting components forming a complex whole, which is delineated by its boundaries, surrounded by its environment, and characterised by its structure and functions (Backlund, 2000; Limburg et al., 2002). However, many socio-hydrological systems have ambiguous boundaries, making it difficult to examine resilience. So, for example referring to 'the system' may mean components or parts of the whole human-water interaction, such as the water subsystem with anthropogenic drivers, or to the human subsystem with hydrological drivers, or to the *socio-hydrological* system, which emphasises the feedbacks and interactions between human and water processes in a balanced and integrated perspective.

Once the system type, or the resilience identity is defined, it helps to answer a series of essential questions that sustain the clarity of the resilience concept and application in socio-hydrological contexts. For example, what aspects of systems are being examined, what key indicators of system state need to be established, what possible desired state is sought, and ultimately what shapes the resilience process (Carpenter et al., 2001; Mao and Richards, 2012). Resilience in this context is mainly driven by two factors – hazards and external conditions, often referred to as 'fast' and 'slow' variables (Walker et al., 2012). Hazards are threats to a system, usually comprising occasional, recurrent and continuous perturbations (Gómez-Baggethun et al., 2012; Kuo et al., 2012; Turner et al., 2003) such as diffuse pollution, land-use change, drought and flooding. External conditions or controlling variables include changing climate altering the influence of legal and socio-cultural contexts, and the role of science and technology on the *stability landscape* which is used here as a metaphor to describe the resilient process of systems (cf. Dent et al., 2002; Scheffer et al., 2001). Systems can shift from one position to another, which can result in large, abrupt, long-lasting changes to their structure and function (Biggs et al., 2009). Resilience management seeks both to reduce hazards to prevent the system shifting to an undesirable position (e.g. degradation of ecosystems and living standards), and to move the system toward a desired position. The stability landscape itself can also alter because of change in slow variables. This resilience process is usually represented as a bifurcation diagram, showing system state as a ball moving between equilibrium positions (Scheffer et al., 2001; see Fig. 1).

Thinking through how *absorptive, adaptive* and *transformative* capacities feature in these diagrams opens up new possibilities for understanding systematic and resilience properties (Walker et al., 2004, 2009). Based on Fig. 1a, the interrelation between three resilience capacities is portrayed in Fig. 1b. *Absorptive capacity* equates to the original concept of resilience: that is, the capacity of absorbing disturbance while retaining essential structures and functions (Cumming et al., 2005; Holling, 1973; Walker et al., 2004). It is represented as the size (e.g. width) of the equilibrium region (Walker et al., 2004). This capacity is closely connected with the notion of sensitivity (McGlade et al., 2008; Yan and Xu, 2010). *Adaptive capacity* is usually defined as the systemic capability to respond to perturbation from a changing environment through adjustment and alteration. If absorptive capacity describes system tolerance to change in structure and function under existing conditions, then adaptive capacity denotes how much this absorptivity can increase in response to external change and to change-induced uncertainties in the future (Engle, 2011; Gallopín, 2006; Smit and Wandel, 2006). It also determines resilience by moving the tipping point and making the desired attraction basins wider or deeper, although it does not necessarily lead to improved system state (Walker et al., 2004, see red dotted line in Fig. 1b). Lastly *transformative capacity*, or transformability also refers to the ability to respond, but in a more radical way. This is a capacity to change the stability landscape or even create a new system by means such as introducing new components or new ways of living, when existing

ecological, economic or social structures are untenable (Folke et al., 2010; Walker et al., 2004). Transformations aim to navigate the systematic transitions from an undesired stability landscape to a new, desired state (Folke et al., 2010, see Fig. 1b; Walker et al., 2009). Proactive transformation may be attempted if change in external conditions is so great that incremental improvement through adaptive capacity is inadequate to meet managerial goals (Béné et al., 2014; Ernstson et al., 2010).

## 3     A typology of human-water couplings and resilience framings

The importance of using resilience in the coupled human-water context is increasingly evident in both academic and public policy fields, ranging from aquatic ecosystem conservation (Khamis et al., 2013; Sala et al., 2000) to hydrological risk management (Adger et al., 2005; Hallegatte et al., 2013), and sustainable water use and development (Pahl-Wostl et al., 2013; Vorosmarty et al., 2000). For the reasons explained already, it is critical to clarify the character of resilience in socio-hydrological contexts which in turn is determined by the human-water coupling type. Hence, we identify three types of human-water couplings with their own resilience framings: (1) the *water subsystem,* with hydrological resilience to anthropogenic hazards; (2) the *human subsystem,* with social resilience to hydrological hazards; and (3) the *social-hydrological system,* with socio-hydrological resilience (Fig. 2). We therefore argue that *socio-hydrological resilience* should refer to *resilience of socio-hydrological systems as a whole,* which is one specific type of resilience in socio-hydrological contexts. These three types represent different perspectives from which to approach and understand socio-hydrological systems and human-water interactions, whilst emphasising that humans and water are fundamentally interrelated. The former two types focus on intrinsic hazard-subsystem relations, while the latter covers these subsystem relations and broader and more iterative interplay between them. Here, it is worth noting that socio-hydrology and hydro-sociology have close connections (Sivakumar, 2012), but different emphases (Wesselink et al., 2016). Here we adopt Sivapalan's interpretation of socio-hydrology which has as its focus the co-evolution and feedbacks of coupled human-water systems (Sivapalan et al., 2012). Thus these three types of coupling encapsulate how different fields (e.g. conservation, disaster management and water resources management) deal with human-water couplings, rather than normative expectations of what people should (or should not) do. In the following subsections, features such as resilience subjects, desired system states, indicators and application fields are examined for each type of resilience framing (Table 1).

### 3.1     Water subsystem with anthropogenic hazards

Resilience was advanced initially as a phenomenon of freshwater ecosystems to explain the dramatic change in aquatic ecosystems precipitated by anthropogenic disturbance (Table 1), such as algal blooms caused by nutrient enrichment and fish population collapses triggered by overharvesting (Holling, 1973). Based on these cases, this type of coupling describes a particular aspect of human-water interactions, which focuses on water subsystems and external anthropogenic factors shaping water subsystems. However, from this perspective the human subsystem and its attendant hydrological hazards are not the main emphasis.

Resilience has since been used in many water subsystems, such as lakes, rivers, and oceans (Dudgeon et al., 2006; e.g. Gibbs, 2009; Hoegh-Guldberg et al., 2008; Willis et al., 2010), where hydrological conditions can be measured by surrogate indicators (e.g. Holling, 1973), or through examining other biotic or abiotic components. This coupling model is primarily used in aquatic ecosystem conservation and management, where the goal is to maintain subsystem equilibrium or restore it to a desired historical state. Climate change or increased human hazards may degrade aquatic ecosystems or propel them to irreversible undesired end-states (O'Reilly et al., 2003; Sala et al., 2000), prompting a need to consider options for resilient water management (Mace, 2014). For example, climate change and ocean acidification together with local anthropogenic stress decrease sea water quality, alter community structure and diversity, change species distribution and might even push ecosystems such as coral reef to functional collapse (Carpenter et al., 2008; Doney et al., 2012; Hoegh-Guldberg and Bruno,

2010). In such cases, aquatic organisms (e.g. macroinvertebrates and macrophytes) can be used for biological monitoring to assess adverse human impacts on species and ecosystems (e.g. Miller et al., 2007; Ozkan et al., 2010). Ecological indicators have been developed for river basin management in many regions of the world (Bunn et al., 2010; Josefsson and Baaner, 2011). Attention has also been paid to resilience of hydrological aspects of water systems under climate change, extreme weather and alteration in land cover (Harder et al., 2015; Peterson et al., 2012). Better understanding of multiple steady hydrological states and the process interaction of switching between states can inform adaptive water management (Botter et al., 2013).

Absorptive capacity of water subsystems is mainly from essential ecosystem and hydrological processes. The adaptive dimension of hydrological resilience usually comes from a diversity of species, habitat or landscape. From a biophysical viewpoint, adaptation refers to the response of organisms to their environment at the genetic, individual and/or ecosystem scale (Engle, 2011; Hoffmann and Sgrò, 2011; Krimbas, 2004). This underlies redundancy and natural selection, which drives evolution (Krimbas, 2004; Lindner et al., 2010). However, the improvement of hydrological adaptive capacity does not exclude anthropogenic contributions, and can be achieved by restoring the biodiversity and integrity of aquatic ecosystems. Transformative capacity is seldom used in this water-subsystem-centred type, because the idea of creating an entirely new system is inconsistent with the philosophy of conservation, which focuses on maintaining the natural world.

## 3.2    Human subsystem with hydrological hazards

The second type of coupling is the human subsystem with hydrological hazards (Table 1). Here hydrological components are not considered as a system, but as adverse impacts on human well-being. Although hydrological hazard may be caused or increased by human activities, or its impacts on human society may be exacerbated by inadequate management or responses (Van Loon et al., 2016), emphasis on resilience from this perspective is on how hydrological hazards affect human subsystems, and how human societies respond to these hazards, rather than how water subsystems are changed by human activities. This human-hydrological coupling is commonly applied in disaster management (Kelman et al., 2015; Sudmeier-Rieux, 2014), where resilience is derived from capacity building within human systems to better cope with more frequent hydrological shocks (for example, those induced by climate change; Adger et al., 2005; Aerts et al., 2014; Dahm, 2014).

Human subsystems have many facets and their state is described through numerous indicators and disciplinary approaches. Similarly, resilience understandings vary widely. Meerow et al (2016) describe human subsystems as complex arrangements of processes and phenomena at many different scales and levels. Reviewing 675 articles on resilience, Ostadtaghizadeh et al. (2015) identify five main domains of human subsystems, including social, economic, institutional, physical and natural categories. For example, hydrological hazards may cause injuries, death, and property and infrastructure loss (Liao, 2012), which can be quantified to estimate the approximate cost of disasters (Keating et al., 2015). Apart from this physical aspect, socio-economic condition can also be used to capture the degree of resilience of human systems to hydrological impacts, with economic growth, incomes and livelihoods often used as proxies (Kumar, 2015; Plummer and Armitage, 2007).

Resilience of human systems is usually evaluated from social science perspectives (Lorenz, 2013; Olsson et al., 2015), through concepts such as social capital and network structures, institutions and power relations. Knowledge and discourses have received increased attention (Keck and Sakdapolrak, 2013; Wyborn, 2015). Cutter et al. (2008, 2010) highlight 'community competence' as capacities to understand risks, promote individual physical and emotional health (Norris et al., 2008), and maintain cultural norms such as livelihood practices and social institutions (Crane, 2010). Indeed, recent studies highlight that an alternative approach to engage with challenges posed by resilience is to use a more theoretically pluralist perspective that enhances engagement and utilisation of insights from different angles, alongside insights gained from resilience scholarship (Karpouzoglou et al., 2016a).

Consequently, resilience from the perspective of managing human subsystems tends to emphasise particular societal expectations in relation to how to deal more holistically with hydrological hazards. If social aspects of human subsystems are also considered, more anticipatory targets can be discerned. However, critics of resilience have argued that there is still significant scope for developing a more nuanced understanding of resilience and how it relates to society. Cote and Nightingale (2012) argue that there is still far less attention to normative and epistemological questions. For example, the policy use of resilience is often normative in the sense that it implies that resilience is always something 'good' to be strived for. However, the tendency to see resilience as being an objectively defined desirable can create challenges for social scientists working with the concept (Olsson et al., 2015). An important point is that questions that relate to power and politics of both how and who gets to define resilience need to be brought into the foreground of resilience research, otherwise resilience runs the risk of becoming a power-blind concept (Davoudi et al., 2012). A similar point is made by MacKinnon and Derickson (2012) in arguing that resilience as a concept is too conservative in outlook, because it embraces contemporary societal expectations rather than challenging them; they go further by advocating a shift from resilience to resourcefulness as a concept that better matches the aims of emancipatory social sciences. West et al. have argued that some of the criticisms around resilience can be overcome through identifying better ways for researchers from social and natural science backgrounds to open new dialogues, so establishing common ground while identifying areas of disagreement (West et al., 2014).

In this type of framing, absorptive capacity is the ability to defend from hydrological hazards, while social adaptive capacity is a means to improve this ability and reduce the vulnerability of human subsystems including individuals, communities, groups and institutions in coping with water related shocks and changes (Bennett et al., 2014). Gupta et al. (2010) reviewed the existing literature and summarise six dimensions of adaptive capacity: variety and diversity of problem framing and solving, learning capacity, room for autonomous change, leadership, resources and fair governance. A similar conclusion is made by Bennett et al. (2014) in their four categories of adaptive capacity including flexibility and diversity, capacity to organise, learning and knowledge, and access to assets. Besides incremental improvements, human subsystems can even radically reorganise communities and proactively transform into entirely new settings under global change. An extreme example is climate change-induced migration; here, the subject of resilience under contingent hydrological impacts (populations at-risk) may abandon settlements, migrate to new locations and restructure human subsystems (Methmann and Oels, 2015).

## 3.3    Socio-hydrological system and its resilience

While it is possible to examine resilience from the perspective of water or human subsystems, we argue that it can also be considered in relation to coupled socio-hydrological systems within which human and water subsystems are constitutive elements. This move to socio-hydrology as a framing device implies the need to reassess resilience from a co-evolving viewpoint, where water and human systems make and remake each other and are interdependent in time and space (Sivapalan et al., 2012), so implicating water and society in governance arrangements (Sivakumar, 2012). Here it is the state of the coupled system rather than a particular perspective of either water or human systems that is of interest. This third type of coupling foregrounds the states, conditions and interactions of coupled human and water subsystems to build a more balanced understanding of their process interrelationships, and highlights resilience of socio-hydrological systems to both internal and external hazards.

As discussed in the previous section, desired states of the water subsystem are usually high naturalness or historical conditions measured by biotic and abiotic indicators, while desired states of the human subsystem are more normative societal expectations set by relevant social groups. However, it is a challenge to define the current state as well as the desired state of this coupling type of human-water system, which helps to clarify the identity of socio-hydrological resilience and to

answer 'resilience of what'. A conventional approach to evaluate coupled systems is to use compositional indicators (Meerow et al., 2016). Components from subsystems are assessed separately and then summed up to obtain a proxy value for the overall coupled system state. For example, disaster resilience index usually regards the overall system as a comprised of constitutive ecosystem and human subsystem domains (i.e. social, economic, institutional, and physical) (Ostadtaghizadeh et al., 2015). However, if used in the socio-hydrological context, this compositional approach cannot gauge the complex interactions and feedbacks of human-water coupling (Montanari et al., 2013). Instead, measures are needed that model the dynamic interdependencies of continually interacting components (Gao et al., 2016). This demands a direct assessment of the coupled system using indicators or measures that depict multi-directional interactions. Examples include human benefits from hydrological systems, water resource use, and water-supported socioeconomic development, governance over water, and societal and behavioural response to hydrological hazards (Carey et al., 2014; Elshafei et al., 2014). Among these we argue that the notion of hydrological ecosystem services, which attempt to bridge the two subsystems, is a promising framework to describe the socio-hydrological state and to be incorporated into the resilience thinking (Biggs et al., 2012, 2015; Engel and Schaefer, 2013). In effect, the level of ecosystem services provision is the product of conflicting factors from both sides, such as human demand and ecosystem supply, human disturbances and ecosystem regulation and regeneration, and human management and water resources.

The possibility that hydrological ecosystem services offer a good proxy of human-water intersections is also reflected by its normative goals. Thus, high ecosystem service provision implicitly requires integration of at least three components. First is healthy biophysical systems. Robust ecosystem structure, processes and functioning are necessary pre-conditions for the sustainable provision of ecosystem services (de Groot et al., 2002). Second is the intrinsic value of biophysical systems to human society, even if the value does not have a direct use (Pearson, 2016). Third is the range of established routeways in human societies to channel benefits from nature. This implies that using ecosystem services to measure the state of socio-hydrological systems not only reflects the 'naturalness' of the hydrological system, but also human preferences for the resulting coupled system (Dufour and Piégay, 2009). So a continuing supply of ecosystem services does not necessarily mean ecosystems are pristine or close to a 'natural' condition, but instead reflects the dependence of the human subsystem to select for particular services (National Research Council, 2013; Zedler, 2000). Ecosystem management thus improves the resilience of ecosystems by deliberate human interventions to achieve a desired level of ecosystem services of a preferred sort.

According to the Millennium Ecosystem Assessment (2005), each water ecosystem provides multiple benefits to human society, including (1) provisioning services such as water, aquatic products and hydropower; (2) regulating services including water purification, flood and climate regulation; and (3) cultural services or nonmaterial benefits obtained from aesthetic or spiritual enrichment, recreation, scientific research and educational activities. Vigerstol and Aukema (2011) identify four processes that produce water-related ecosystem services – water retention, water yield, natural water filtration, and water quality purification. Terrado et al. (2014) specify four hydrological ecosystem services vulnerable to climate extremes – drinking water, hydropower production, nutrient retention and erosion control. Fisheries and products from aquatic ecosystems are essential for human societies but also subject to change, and need to be sustainable and resilient (Barange et al., 2014). Ecosystem services as a framework therefore link the human and water system, while being a viable basis for decision and policymaking (Brauman et al., 2007; Daily et al., 2009). Thus, managing socio-hydrological resilience can be understood as regulating and enhancing resilience of ecosystem services that support livelihoods and human needs for natural hazard protection, making it a viable proxy for socio-hydrological systems.

Resilience of socio-hydrological system may not only come from its water or human subsystems, but from human-water interactions that are not prominent in the first two types. For example, real-time monitoring of hydrological disasters contributes to absorptive capacity. Adaptive capacity can be underpinned by water governance and institutions, as well as

environmental knowledge learning and exchange. Transformative capacity may be rooted in the incentive, ability and innovation in optimisation of water usage model, development of water-dependent socio-economic structure, and reconstruction of human-water relations through resettlement (Arnall, 2015; Barrett and Constas, 2014; Wilson et al., 2013). Identifying and analysing the sources of capacities helps to design resilience building strategies for socio-hydrological systems in a more comprehensive way, without missing valuable improvement possibilities.

# 4    Pathways to resilience in the socio-hydrological context

Building on the preceding section, here we conceptualise resilience in the socio-hydrological context as a normative goal that can be achieved through human intervention.

## 4.1    Resilience capacities and the 'resilience canvas'

Building resilience requires not only improvement of the absorptive capacity to resist existing hazards, but also enhancing system resilience to cope with future uncertainties. This is where the properties of *adaptive* and *transformative* capacity advanced here enrich the socio-hydrological perspective. By conceptualising resilience this way, represented by increased *adaptive* and *transformative* capacities, the need for incremental adjustment or radical improvement of systematic states becomes clearer.

An analogy can be drawn with conservation ecology. Gillson et al. (2013) use two axes of concerns (landscape vulnerability and conservation capacity) to design conservation strategy. Based on this approach, here we introduce the 'resilience canvas' by combining two of the constitutive capacities as the x- and y-axes (Fig. 3). This section demonstrates how the 'resilience canvas' can be constructed and applied, by emphasising on the first two dimensions of resilience – absorptive capacity for current hazards and adaptive capacity for future contingencies. The transformative capacity is not focused in the discussion because it requires some further exploration compared to the first two capacities – there is still an ongoing debate on what exact systematic attributes are needed to support a radical transformation to an entirely new stage (Robinson and Carson, 2015; Wilson et al., 2013). Here we keep the analysis of resilience capacities in a visually simple way as a 2-dimentional space instead of a 'resilience cube', and select the first two  capacities for demonstration purposes.

Four resulting system states are found at the corners of the canvas: most resilient (top-right: high absorptive and high adaptive), vulnerable (bottom-left: low absorptive and low adaptive), susceptible (top-left: low absorptive and high adaptive) and resistant (bottom-right: high absorptive and low adaptive). These four quadrants are not static, and systems can move between them via structured management interventions over time, which we term 'pathways'. A resilient-vulnerable gradient from top-right to bottom-left is shown on the canvas (Fig. 3).

## 4.2    Building pathways to resilience in socio-hydrological contexts

The pathways on the resilience canvas represent a series of three hypothesised human intervention scenarios introduced to effect system change (cf. Haasnoot et al., 2013) (See also Fig. 3). These are hypothesised in the sense that in adopting a broad definition of resilience, these pathways could be very different depending on the social actors and hydro-social context of operation However, for the purposes of illustrating how the pathways approach could be useful in the case of the resilience canvas, pathways help steer socio-hydrological systems towards the 'most resilient' status (i.e. top-right of the canvas).  This is regarded for the purposes of this study as the most valued water management goal.

Susceptible socio-hydrological systems can be strengthened by increasing absorptive capacity, and by making hydrological ecosystem services supply more robust and sustainable under current hazard regimes. For example, water pollution may decrease potable water availability, while introducing vegetated buffer zones can protect water quality(Hickey and Doran,

2004; Khamis et al., 2013); aquatic ecosystem degradation may shrink fish populations and food yield from aquatic products, and diversifying abiotic characteristics such as habitat supports the resilience of faunal populations (Bisson et al., 2009; Khamis et al., 2013). Hydrological disasters also deplete human benefits derived from water systems, and setting up early warning systems can increase substantially the capabilities to deal with disasters (Adger et al., 2005).

By contrast, for resistant systems approaches are needed to improve system adaptability and capability to cope with future disturbance. Adaptive capacity can be enhanced in several ways. One approach is to restore the essential ecosystem processes that generate services. For example, hydrological adaptive capacity depends on various intrinsic factors such as biomass, biodiversity and ecological traits of species (Dawson et al., 2011; van Vliet et al., 2013). In an abiotic context, adaptive capacity can also be determined by features such as high river connectivity (Khamis et al., 2013), stable
hydrological cycles (Thomas, 2016), and heterogeneous landscape (Czucz et al., 2011). A second approach is to raise social and institutional capabilities, such as accessibility to information and resources (Milman and Short, 2008), responsiveness to environmental change (Malhotra et al., 2007), enhance institutional structure and governance processes (Folke et al., 2005; da Silveira and Richards, 2013), boost stakeholder participation (FEW et al., 2007), and encourage learning and knowledge dissemination and exchange (Pahl-Wostl, 2009).

Although pathways can be constructed for the four system states, factors that improve different capacities via different capacity sources (i.e. ecological/ hydrological, social or human-water interactions) cannot always be distinguished or promoted independently. For example, maintaining diversity and redundancy of system components such as species, landscape types, knowledge systems, actors, cultural groups and institutions, benefits systematic resilience in various ways; so managing connectivity not only facilitates system recovery, but also improves the responsive capability to future
uncertainties (Biggs et al., 2012, 2015). These activities are applicable for both social and natural sciences, and cut across the three socio-hydrological resilience capacities.

Scheffer et al. (2015, p.1317) suggest keeping systems within the 'safe operating space' by managing down local stressors (fast variables) to a low value and responding to future climate projections (slow variables). The 'resilience canvas' portrays this management strategy from the perspective of preparedness instead of driving variables. The impact of local stressors as
well as climate change can be better mitigated with increasing absorptive and adaptive capacity respectively. It implies that social-hydrological systems should not only be kept within predetermined operating limits but also be the focus of bespoke resilient strategies. Khamis et al. (2013) compare the network sensitivity and conservation capacity of two catchments – the Taillon Catchment in French Pyrénées and the Rhone catchment in Swiss Alps – by assessing nine variables. It was found that the Rhone catchment has relatively higher absorptive capacity because of its lower network sensitivity, lower potential
for alien species invasion, and higher cryosphere-flow buffering, while the Taillon catchment has higher adaptive capacity due to its larger proportion of conservation area and higher naturalness of river flow. Overall, the two catchments have similar evaluation of resilience for their similar distance to the 'most resilient' stage on the 'resilience canvas' (Fig. 4). However, customised strategies should be developed for each catchment to achieve the resilient goal.

## 4.3    Resilience trajectory of global socio-hydrological systems

The resilience canvas can be used at scales from the river basin to the global. By analysing our preparedness to cope with local stressors and change, the resilience canvas illustrates a development trajectory for global socio-hydrological systems (Fig. 5). This section suggests that pathways are not always in straight lines, while the constitutive capacities of resilience do not usually grow equally while the overall resilience is increasing. It also shows the potential to shift from resistant to resilient water management strategies, and on this basis, identifies attendant future research and implementation gaps.

The development phases of global human-water relations are identified and discussed in the literature. Mace (2014) for example argues that we are experiencing a shift in emphasis from 'Nature suffering from People' or 'People benefiting from

Nature' to 'People and Nature' as a more interdisciplinary and interactive framing for conservation purposes. Gleick et al. (2009; 2010) also identify three water eras characterised by contrasting water challenges and problems as follows: nature's water resources; intensively manipulating water sources; and massive global crisis, a stage demanding interdisciplinary and integrated approaches for management purposes. By examining the history of the Murrumbidgee River basin, Australia, Kandasamy et al. (2014) recognise four main development eras of socio-hydrological systems: building irrigation and associated infrastructure; gradual appearance of environmental degradation; awareness of environmental impacts and application of consensus strategies; and switching to a directed government interventionist strategy. There is a lagging societal and governmental response to environmental change during the development – it can take years to aware the side-effect of infrastructure construction, and to test and perform the remedial measures until they have an effect. Kandasamy et al. described this changing attitude in respect of the environment as a 'pendulum swing' – the balance point in water allocations is turning around between humans and ecosystems. Along with the development of socio-hydrological systems, their resilience changes and evolves simultaneously. Therefore, based on these classifications, three main stages of socio-hydrological *resilience* at the global scale are summarised and presented on the resilience canvas (Fig. 5).

1. *People with Water*. Before intensive modification of environments, human societies mainly relied on natural hydrological cycles for subsistence (e.g. hunter gathering), and to support extensive low productivity agriculture reliant on limited control of the water subsystem (Gleick, 2009). The ecosystem services were often vulnerable to internal or external hazards, such as water-related diseases and adverse hydrological events, because of insufficient physical and institutional preparedness. However, the unoptimized ecosystem services did not cause too many problems, because of the small and dispersed population and low demand on hydrological ecosystem services (Gleick, 2009). Therefore, at this stage, absorptive capacity was low and adaptability was mainly provided by the 'naturalness' of ecosystems.

2. *Water for People*. As population has grown and socio-economic development risen, ecosystem services obtained by human societies from water systems have no longer proved sufficient. New technologies and approaches were invented to intentionally manipulate water cycles to meet new societal demands (Gleick, 2009). Along with the development of hydraulic engineering, humans have dramatically increased the range of ecosystem services obtained from water subsystems, such as hydropower, water availability, flood regulation and more intensive forms of food production. This development of socio-hydrological systems explicitly emphasised the benefits people received from water, and marks a transition to a 'Water for People' framing (Mace, 2014).

The resilience of socio-hydrological systems was also altered according to prevailing socio-economic and cultural conditions. So, river canalisation increased the absorptive capacity to flooding, but at the cost of rapid water transfer downstream. Major water transportation projects were built to transfer water as well as ecosystem services from wet to dry areas, in order to increase socioeconomic resilience (Langridge et al., 2006). Hydraulic engineering was conducted to increase the system's absorptive capacity to cope with existing and known hazards (e.g. flooding, drought and pollution). For example, damming was once regarded as one of the best solutions to avoid flooding and drought (Endfield, 2012; Ward, 2005). However, this improvement was at the expense of natural ecosystems, and consequently decreased ecological adaptive capacity.

3. *People and Water*. Despite increasing capacities to manage water, societies face water supply crises amid growing realisation that climate and global change are making this ever harder to address. For example, climate change may alter the prevailing hazard regime and put ecosystem services at risk, though it is not straightforward to know what the new regime and risks are. Thus, canalised rivers do not have the surplus capacity to absorb more frequent rain events or higher surface runoff, which leads to more severe flooding. Fernald et al. (2015) discover that the traditional acequia systems may not be still tenable, when external drivers brought by climate and land cover change push these systems

beyond their historical limits. The spatial distribution of precipitation regimes may also be shifting under climate change, and this may turn regions from humid to dry, or vice versa (Collins et al., 2010), making water transportation projects redundant. These hard-engineering approaches generally have less flexibility and usually have a lengthier time lag in responding to change. The socio-hydrological system at this stage acts as a valuable heuristic for adaptive water management, offering the most resilient hydrological ecosystem services supply. The 'People and Water' framing has shifted from the linear one-way relationship of 'Water for People', to a multi-layered and multi-dimensional relationship between human societies and water systems (Mace, 2014).

We are now facing a new challenge of future water contingencies and uncertain water-related hazards, which was transited from the historical challenge of meeting growing needs of hydrological ecosystem services. It implies that a shift of water strategies is urgently required. We argue that most current water management practice is now seeking to transition from resistant to resilient strategies (Gillson et al., 2013; Khamis et al., 2013) (Fig. 5). From the perspective of a resilient socio-hydrological system, we believe that this strategy is not only needed for water management, but potentially offers a feasible alternative for achieving sustainable hydrological ecosystem service provision. Awareness of change suggests an increase of adaptive capacity, and implies that the pathway to socio-hydrological resilience will involve 'soft' approaches that are complementary to engineering-based methods (Park et al., 2013). This implies that interventions in all the source of resilience, including water and human subsystems and human-water interactions, need to be considered. Some promising example approaches are provided as follows. Ecosystem restorations such as decanalisation, improving river connectivity, and floodplain recovery also suggest enhancing ecological or hydrological adaptability (Brauman et al., 2007), although in the process absorptive capacity may be compromised (Chen et al., 2016). This reemphasis on the ecosystem integrity in response to degrading environmental quality is also aligned with the 'pendulum swing' phenomenon discussed by Kandasamy et al. (2014). Polycentric water governance and public participation in more centralised forms of decision-making may play important roles in building socio-hydrological resilience (Buytaert et al., 2014, 2016). Polycentric systems have multiple governance units at multiple scales, which provide the flexibility to deal with the target problems at appropriate scale, and offers institutional back-ups to respond to uncertainties (Andersson and Ostrom, 2008; Garmestani and Benson, 2013). In addition, technological innovations as well as advances in data collection and prediction models also contribute to improving socio-hydrological resilience, in both absorptive and adaptive ways (Karpouzoglou et al., 2016b). For example, the Environment Agency for England and Wales offers early warning systems that provide forecasting and personalised household information, which builds upon developing technologies and skills to map and measure risk (Environment Agency, 2009).

# 5     Concluding remarks

Evaluating resilience in a socio-hydrological context is challenging because of different framings of water-related resilience, including hydrological resilience to anthropogenic disturbances, social resilience to hydrological disasters, and socio-hydrological resilience. Although these reflect different aspects of human-water interactions, they are not easy to distinguish. To better conceptualise the linkage between resilience and socio-hydrology, we have emphasised the need to define the system type prior to discussing their intrinsic resilience, and have argued that resilience be regarded as a set of systematic properties including absorptive, adaptive and transformative capacities. Based on this understanding, we have proposed a conceptual framework of human-water couplings and resilience framings, including a heuristic approach to identify possible pathways to resilience in socio-hydrological contexts.

Focusing on three coupling and framing types, we highlighted the potential of socio-hydrological resilience. If human societies are considered as endogenous components of water cycles, this newly proposed resilience concept is useful to answer how social-hydrological systems respond to and cope with perturbations. On this basis, we have shown the utility and

complementarity of resilience with ecosystem services, and argued that the framework of ecosystem services can be a promising tool to describe the resilient dynamics of socio-hydrological systems, since it reflects an essential aspect of the human-water interface.

Different types of resilience may match particular problems with knowledge and research traditions in certain academic fields. For example, hydrological resilience to human hazards may be usefully analysed with biophysical sciences for aquatic ecosystem conservation, while social resilience to hydrological hazards will require significant inputs from social sciences. It is important to consider the different nature of human and biophysical systems where different resilience approaches are used. For example, the ecologically-oriented concept of resilience has received critiques when applied in human systems, because it oversimplifies the understanding of equilibria and feedbacks, ignores the importance of social conflict and power, and addresses the notion of system function which is not the key focus in social science (Olsson et al., 2015). This does not mean that resilience should be discarded as a concept. However, we should heed calls for pluralism, stimulate dialogue and develop a clearer identity of resilience as applied in the socio-hydrological context (Cote and Nightingale, 2012; Cumming et al., 2005; Olsson et al., 2015).

Resilience is not only a descriptive notion, and usually has normative (goal-setting) objectives. To build pathways to socio-hydrological resilience, we introduced the notion of the 'resilience canvas' to compare absorptive and adaptive capacities. The resilience canvas can be used to design bespoke interventions and strategies for all types of human-water couplings at different scales from single river basin to global level. On this canvas, we showed that the global socio-hydrological system has moved from the stage in 'People with Water', through the 'Water for People' stage, towards the 'People and Water' stage, along with people's growing demand on water and the increasing resilience of hydrological ecosystem services supply. Nonetheless there is still substantial geographic variation globally in the distribution of these socio-hydrological stages.

Therefore, this new conceptual framework with the 'resilience canvas' motivates some future work on resilience. For example, we need to review, compare and classify existing resilience indicators, propose new quantification and assessment methods for different resilience framings, or even develop mathematical tools to quantitatively describe the resilient processes of the capacities (Gao et al., 2016). We also need to conceptualise resilience dynamics and pathways over time with empirical studies, and shift resilience studies from focusing on single cases at particular points in time, to macro-scale comparisons between the past, present and future. In addition, our argument provokes a rethinking of using resilience in other human-nature contexts, such as social-ecological systems (Ostrom, 2009), and coupled human and natural systems (Liu et al., 2007). We suggest there is considerable potential to scrutinise the concept of resilience and better refine its identity and capacities in these systems. Similarly, different framings such as ecological resilience, social resilience to ecological hazards and social-ecological resilience can also be recognised; and within each framing, resilience as systematic properties can be viewed at the capacity level by using the resilience canvas. Hence, we argue this conceptual framework can be used to guide and construct discourses of resilience in the human-nature context, so bringing greater conceptual rigor and clarity to bear on one of the most pressing contemporary public policy challenges of our time.

## Author contribution

F. Mao initialised the ideas of the paper with D. Hannah and J. Clark. F. Mao prepared the manuscript with contributions from all co-authors. Figures were prepared by F. Mao.

## Acknowledgements

This paper was developed within the 'Mountain-EVO: Towards a virtual observatory for ecosystem services and poverty alleviation' project, which is supported by the ESPA Project Framework grant (Project code: NE/K010239-1). We thank all the project collaborators from Nepal, Peru, Kyrgyzstan and Ethiopia, for fruitful discussions that facilitated the conception of ideas discussed herewith. We thank Dr Kieran Khamis for his advice in selecting examples for the 'resilience canvas'. There are no additional data associated with this conceptual paper.

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

# Figures and Table

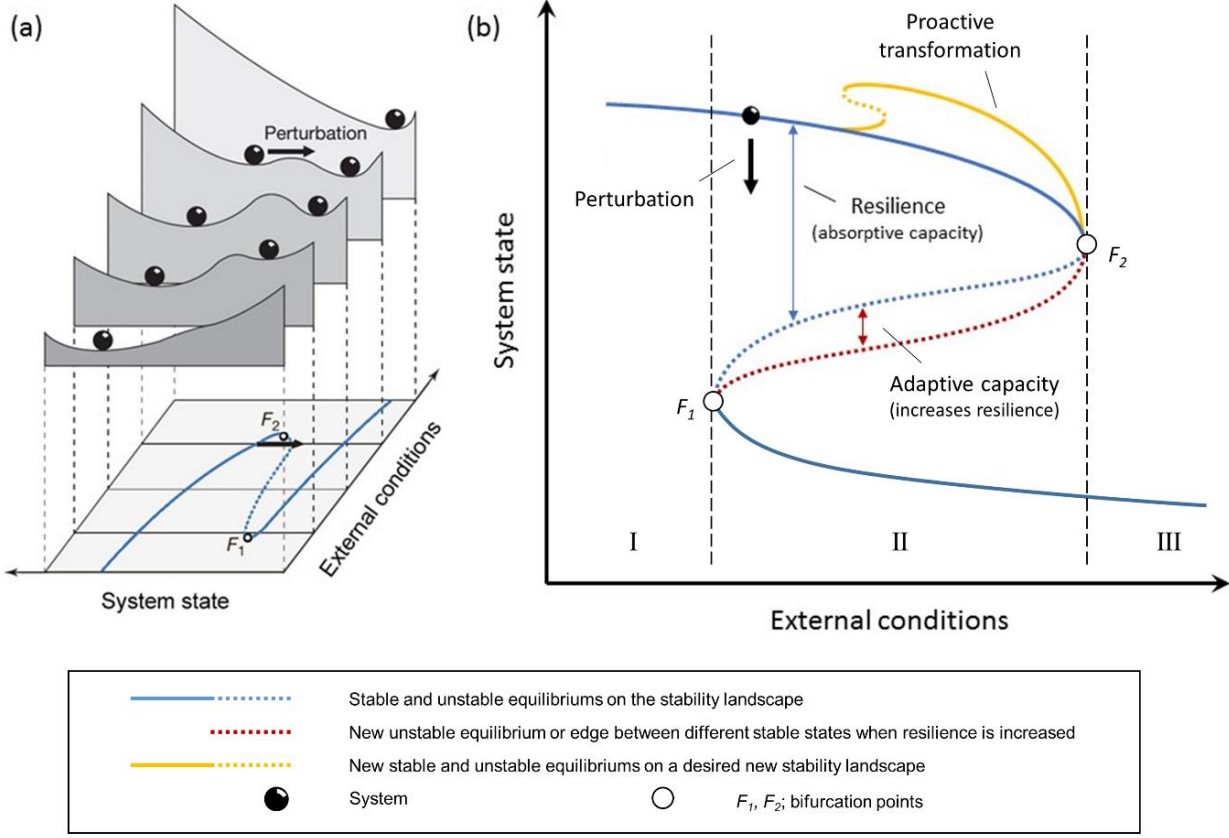

**Figure 1. Bifurcation diagrams for socio-hydrological resilience: (a) Stability landscape changes along with external conditions, adapted from Scheffer et al. (2001). (b) Bifurcation diagram illustrating *absorptive, adaptive* and *transformative* capacities. The two dashed lines across F₁ and F₂ divide the bifurcation process into three phases. Phase I and III have only one stable equilibrium, while Phase II has two stable equilibriums. Perturbations in Phase II may drive the system from one stable state to another. Absorptive capacity refers to the original meaning of resilience; adaptive capacity refers to the capability to increase resilience in response to external change (red line); transformative capacity refers to the capacity to respond more radically, such as proactively navigating the system to a desired new stability landscape (yellow line).**

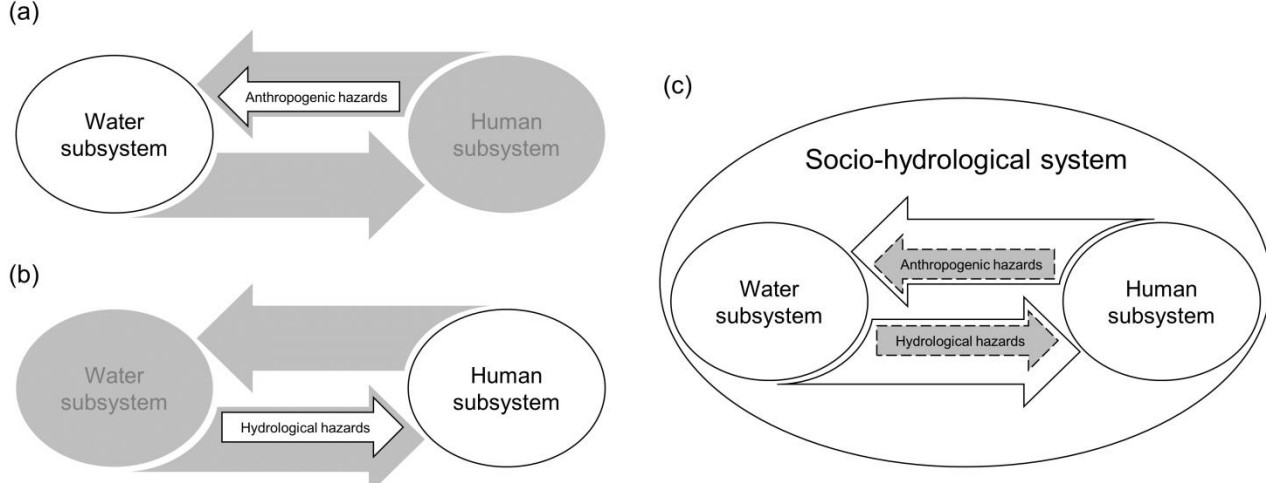

**Figure 2. Schematic diagram of three human-water coupling types, each foregrounding different aspects of socio-hydrological system. (a) Water subsystem with anthropogenic hazards, where the human subsystem, water impacts on human subsystem or other forms of human impacts on the water subsystem are not the main focus. (b) Human subsystem with hydrological hazards, where similarly water subsystem, human impacts on water subsystem or other forms of water impact on the water subsystem are not emphasised. (c) Social-hydrological system with water and human subsystems, and anthropogenic and hydrological hazards as two of many forms of human-water interactions.**

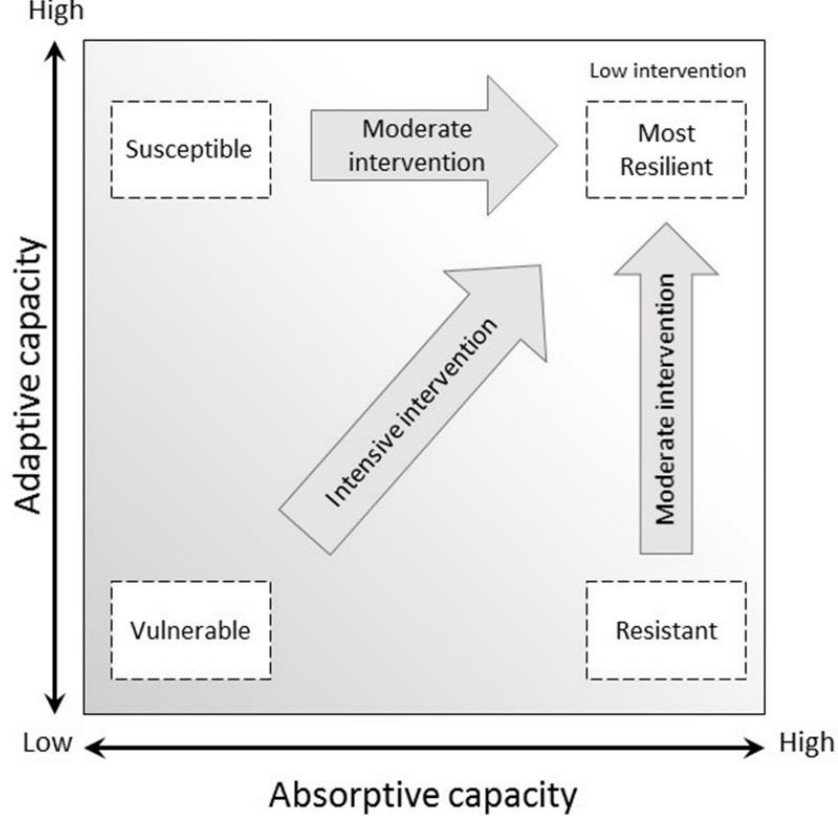

**Figure 3. 'Resilience canvas' with absorptive and adaptive capacities as two axes, showing resilient-vulnerable gradient, resilience conditions and pathways to resilience. The four dashed rectangles illustrate resilient, resistant, susceptible and vulnerable system conditions. The grey arrows represent pathways, or a series of concerted interventions designed to drive systems from one condition to another.**

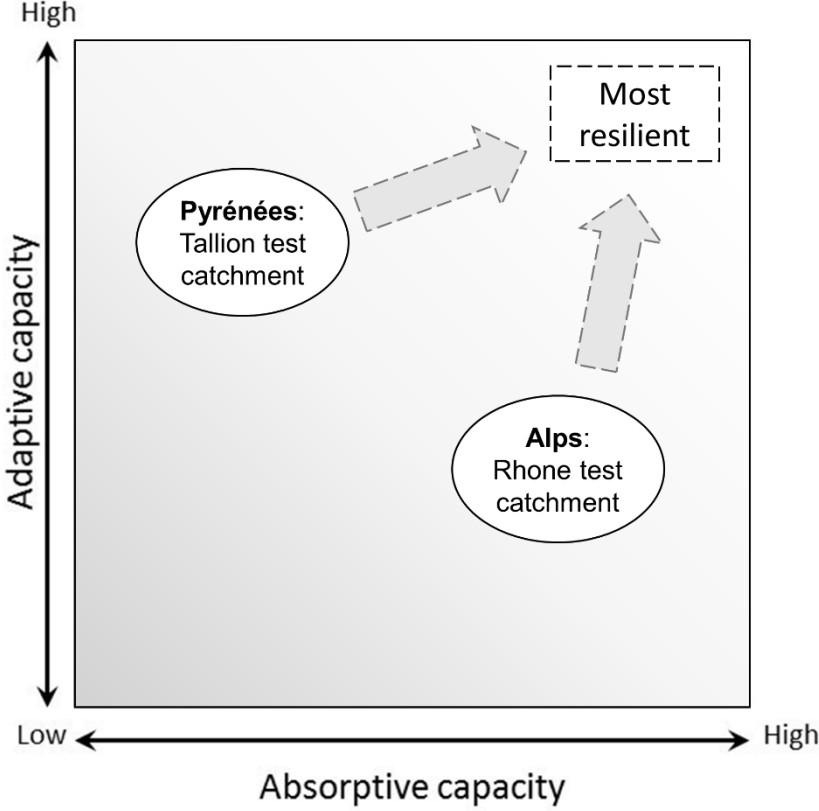

**Figure 4. Regional comparisons of hydrological resilience. Two test catchments are compared, including Taillon catchment in French Pyrénées and the Rhone catchment in Swiss Alps.**

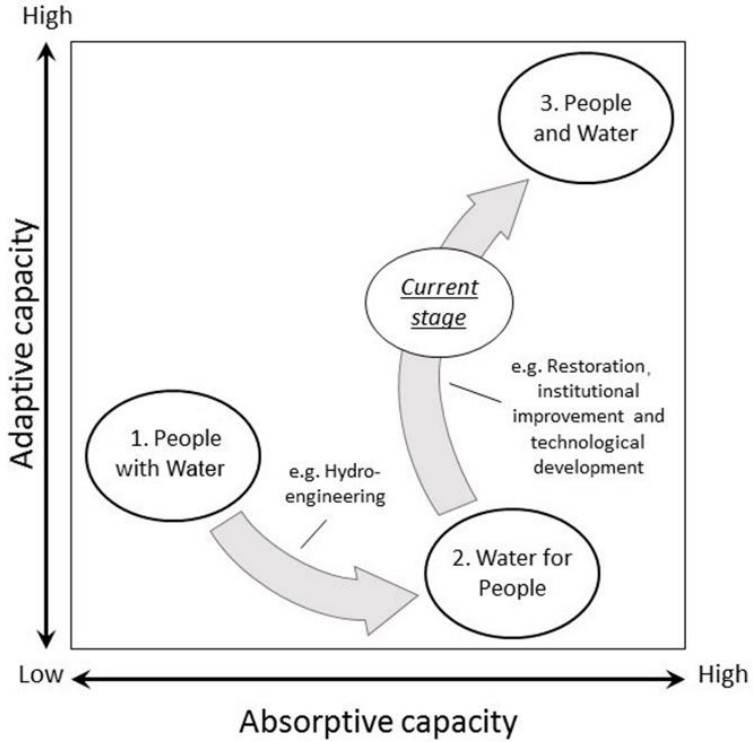

**Figure 5. Global development of socio-hydrological resilience on the 'resilience canvas'. Three main developmental stages are (1) People with Water, (2) Water for People and (3) People and Water. The current global socio-hydrological system has been moving from the second to the third stage.**

**Table 1. A comparison of three types of human-water couplings and resilience framings.**

| Human-water coupling types | Water subsystem with anthropogenic hazards | Human subsystem with hydrological hazards | Socio-hydrological system |
|---|---|---|---|
| System | Water subsystem | Human subsystem | Socio-hydrological system |
| Desired system state | e.g. High naturalness or historical state | e.g. Social prosperity, development and justice | e.g. System integrity, and healthy human-water relationship |
| System indicator | Biotic and abiotic indicators, such as aquatic ecological composition, biodiversity and flow regime | e.g. Social, economic, institutional, physical aspects of human societies | e.g. Compositional indicator and human-water relationship |
| Resilience | e.g. Hydrological resilience and hydro-ecological resilience | e.g. Social resilience | Socio-hydrological resilience |
| Hazards | Anthropogenic hazards | Hydrological hazards | Internal and external, anthropogenic and hydrological hazards |
| Application fields | e.g. Water conservation and restoration | e.g. Disaster management | e.g. Water resources and ecosystem services management |