# Peer review of "A conceptual framework for assessing socio-hydrological resilience under change"

_Hydrology and Earth System Sciences, 2016_

## Referee Comment (RC1) · A. Wesselink (Referee) · 7 Nov 2016

With this paper, the authors aim to clarify and specify what resilience means, or could mean if the potential for multiple meanings is assumed, in a context of human-water systems, as studied by socio-hydrology. They want to provide clarity in the conceptual understanding of resilience so the concept becomes more useful and useable. They draw on existing literature from Socio-Ecological Systems modelling (SES) and complexity science to develop a framework for classification of the state of the human-water system under consideration, which they label 'resilience canvas'. This, they assert, helps to describe historical, global trends in human-water relationships, as well as more 'local' developments e.g. towards sustainable flood management. As I have argued elsewhere (Wesselink et al., in press), socio-hydrology is a recent branch of SES so this foundation seems defendable. To create clarity on the concept of resilience also

seems a good idea; however, I fear that for me this paper does not achieve this, it even creates more confusion. I hope the authors can fix the problems I identify below.

This paper can be assessed in three ways: 1. It's compliance with the orthodoxy, traditions and applications of resilience within SES. Since this area has a much longer pedigree, much can be learnt about resilience from SES research. 2. The internal logic of the paper and its application of SES basic principles. 3. The potential usefulness of the presented work for the application of resilience principles in policy making. The authors avoid being judged on the first criterion when they state (p.3) 'Our aim here is not to describe this variety' [note: they then state they want to instead 'characterise how resilience is interpreted'. How exactly is 'describing variety' different from characterising interpretations?]. I believe they thereby miss an obvious chance to learn from others' work, but it is indeed not possible to present an extensive literature review on resilience in SES literature in the current article. I hope they will publish such an extensive review elsewhere. I have therefore focussed on the second criterion, with some comments on the third.

My first major problem is that the authors state that (p. 3) 'Resilience can be understood as a set of systemic absorptive, adaptive and transformative capacities (Walker et al., 2004, 2009)'. I do not need to go beyond reading the titles of the two references to know that Walker used the concepts 'resilience, adaptability and transformability'. So instead of resilience as one of three system characteristics (as Walker does), the authors use resilience as overarching concept that includes the other two characteristics, and 'absorptive capacity equates to the original concept of resilience' (p.4). They thereby re-define a very well-known concept. This is never a good idea, and certainly not in an article whose main theme is 'conceptual clarity'. I therefore wonder why the authors redefined resilience to mean 'absorptive, adaptive and transformative capacities', what their relationships are, and how they compose together resilience? Re-definitions lead to confusion rather than clarity. Since I am not an expert on resilience studies, I cannot estimate what this re-definition does in terms of changing what is studied and on what

terms. I hope a reviewer with more knowledge in the SES-resilience domain can shed light on this.

I think the treatment of the meaning of resilience in the one way views on human-water systems (sections 3.1 and 3.2) is rather brief. I would especially like to see the application pf the three elements of resilience that the authors defined, i.e. absorptive, adaptive and transformative capacities. In fact, in 31 the authors themselves the 'old' understanding of resilience when they discuss the water subsystem with anthropogenic hazards when they state that (p.5) 'the implicit goal of maintaining subsystem equilibrium or restoring it to a desired historical state' leads to 'resilience management' – surely in the authors' definitions this should read 'managing absorptive capacity'? In section 3.2 the same applies, as again in section 4.3.

Related to the above, why does transformative capacity not appear in the 'resilience canvas'? Is this maybe a possibility for coping that is beyond or against the authors' own norms and aspirations? If this is the case, it needs to be made explicit; however, the internal logic of the theoretical argument loses much strength: first resilience is three-part, then only two parts are used to assess resilience (absorptive and adaptive capacity).

My second major problem is the substitution of 'ecosystem services' for 'resilience' when assessing two-way interactions in human-water systems. In the justification that is presented, ecosystem services are clearly another way to look from the human to the natural system (p.8): 'a continuing supply of ecosystem services does not necessarily mean ecosystems are pristine or close to a 'natural' condition, but instead reflects the preference of the human subsystem to select for particular services'. It turns out that ecosystem services are hardly needed in the rest of the paper, so I suggest to remove it. However, this does leave one of the goals of the paper unfulfilled: to characterise the resilience of the human-water systems. In my view, a major re-think is needed here, or a reduction of the ambitions of the paper. The latter would be perfectly acceptable, since the conceptual content is considerable, as expressed especially in the figures.

My final comments relate to the usefulness of the paper/the resilience canvas for policy making. One element of this discussion relates to the (perceived by the authors) need to specify what resilience means. This is commonly known as an activity of framing or structuring, with resilience classified as wicked or unstructured problem (Hisschemoller & Hoppe 1995; Hisschemoller et al 2001; Hoppe 2008). The authors seem to be aware of this (p.13): 'Resilience is not only a descriptive notion, and usually has normative (goal-setting) objectives'; also in section 3.2 'Resilience from the perspective of managing human subsystems thus emphasises particular societal norms and goals or normative aspirations in relation to hydrological hazards'. At the same time, they do not question the notion of 'desired state': desired by whom? In the conduct of research, it will be the researchers who decide what is desired (often based on ecological arguments). In other words: targets for the water system would be just as dependent on societal norms and aspirations!

Problem structuring is done when resilience is used within research, but even more so when it is stated as a policy objective, since it can be expected that stakeholders' values and interests diverge more than researchers'. I therefore question the notion that researchers should prescribe (i.e. pre-structure) how policy processes should engage with boundary or nirvana concepts like resilience – this should be done in the policy process. In fact, the literature on boundary objects (Molle 2008; Walker & Shove 2007) suggests that interpretative flexibility of goals is essential in policy making.

While I am fascinated by the way in which the authors depict pathways on the resilience canvas, I find the discussion in section 4.2 too general and too obvious. Neither the resilience canvas (or its derivation) nor the concept of socio-hydrology is necessary to make these general statements. Section 4.3 is again interesting [note: this is the only place ecosystems services make a brief appearance, not just as needing to be resilient but also sustainable (p.11 line 25): mentioning this term opens another can of worms at least as large as the resilience confusion!]. However, I do disagree that everywhere the current pathway can be described as 'people & water'. For example, the EA for

[Figure]

England & Wales [NB: not UK!] approach to flood risk management in fact increases individual people's vulnerability by transferring responsibility to individuals rather than the EA.

References Hisschemöller M, Hoppe R (1995) Coping with intractable controversies: The case for problem structuring in policy design and analysis. Knowledge and Policy 8(4) 40-60 available at https://works.bepress.com/robert_hoppe1/7/

Hisschemöller M, Hoppe R, Groenewegen P and Midden C J H (2001) Knowledge use and political choice: a problem-structuring perspective on real life experiments in extended peer review. In: Hisschemöller M, Hoppe R, Dunn W N and Ravetz J R (eds) (2001) Knowledge, power, and participation in environmental policy analysis Policy studies review annual 12. Transaction, New Brunswick

Hoppe R (2010) The Governance of Problems. Puzzling, Powering, Participation. Bristol: Policy Press

Molle F (2008) Nirvana concepts, narratives and policy models: Insight from the water sector. Water Alternatives 1(1): 131‐156

Walker G, Shove E (2007): Ambivalence, Sustainability and the Governance of Socio-Technical Transitions. Journal of Environmental Policy & Planning, 9(3-4) 213-225

Wesselink A, Kooij M, Warner J (in press) Socio-hydrology and hydrosocial analysis: towards dialogues across disciplines. Wiley Interdisciplinary Reviews Water

---

## Referee Comment (RC2) · Anonymous Referee #2 · 28 Nov 2016

Summary: The authors propose a conceptual framework for assessing socio-hydrological system (SHS) resilience under change. It is based on distinguishing 3 resilience concepts, 2 of which are based on the hydro and the socio- sub-system of a socio-hydrological system while the third one focuses on the interactions between the two. The authors propose to understand resilience in terms of absorptive, adaptive and transformative capacities of a system. The authors also develop the notion of resilience as a way to identify SHS pathways to resilience states.

Comments: This is an interesting perspective on conceptualizing resilience of SHS. The concept is almost entirely borrowed from the broad literature of socio-ecological systems and eco-hydrology (wherever applicable). However, this are amenable to the

nature of the science of socio-hydrology, which is open to multiple interpretations of the same phenomenon. This paper is therefore a valuable contribution to socio-hydrology. Below are my main concerns.

1) The authors appear to treat hydro-sociology and socio-hydrology as if these two fields are similar in their demand to understand resilience (or possible instability) in coupled human water systems. The concept proposed by the authors is more formal and requires a post-positivist approach, which seeks regularities and generalizable relationships between human and their water system contingent on disciplines involved. It is not clear where hydro-sociology stands in this context, if it is anti-positivist (i.e. not seeks generalizable relationships) then it will be difficult to interpret outcomes of non-linear system dynamics such as instability and resilience. Socio-hydrology offers not only a quantitative framework but such a framework that allows for bi-directional feedbacks. I wonder if hydro-sociology is more oriented towards impact assessment and implementation of social objectives or outcomes – and bi-directional feedbacks may be key to understanding resilience of SHS. The authors may want to clarify the positioning and capacity of hydro-sociology in being able to implement the formal concept of resilience that is being presented here.

2) Why should we consider the resilience and stability of the hydro and socio sub-systems of a SHS if it doesn't affect the bi-directional feedbacks. It runs very contrary to the philosophy of socio-hydrology which emphasizes that endogenization of human agency as its key consideration. Why should we even be bothered to study instabilities isolated to certain sub-systems when it does not spread to the larger system through its coupled dynamics? Why should we study instabilities in hydrological systems when they are of no concern to humans – current or future! If such instabilities are of concern to humans then we are talking about resilience of the entire SHS, not isolated subsystems. I therefore encourage the authors to phrase all 3 types of resilience given in section 2 in context of the larger SHS and to not isolate them.

3) Same as in point 2, the notion that hazard such as pollution (section 3.1) is a shortterm exogenous perturbation is incorrect in my opinion. It is caused by humans, and human respond to that through community sensitivity and change their norms. Thus hazard is not an exogenous perturbation but an endogenous one as a result of human agency. It is also not a short term perturbation since its effect through community sensitivity can be over decades, even centuries. See for van Emmerik et al. (2014, HESS) and Kandasamy et al. (2014, HESS). I therefore think isolating the concept of resilience to hydro and socio sub-systems is contrary to socio-hydrology and misguided.

4) The discussion of absorptive, adaptive and transformative capacity is not clear. I am surprised that the authors decided to discuss such formal concepts in a qualitative manner. To ensure tractability of concepts, I would encourage authors to present a mathematical toy problem and express these concepts more formally.

5) Page 6, Line 27, "..resilience is to use a more theoretically pluralist perspective..": totally agree, and socio-hydrology welcomes such a perspective.

6) Section 3.2, line 3, page 7 "..  argue that the dynamics of social change should be better framed as part of socio-ecological research.": if you extrapolate it to socio-hydrological systems, then this is more about coupling and bi-directional feedbacks between human and their water system, contrary to the idea of the section that resilience should be treated in (social sub-system) isolation with hydrological hazards as (exogenous) boundary conditions.

7) When I come to section 3, it is not clear where the authors stand in terms of how resilience should be studied (whether in isolation or not). They evoke arguments against the compositional approach but their resilience framing appears to be compositional in nature! The authors should clarify this, I hope in favor of abandoning studying resilience of sub-systems.

8) Lines 3-4, page 8: "..human preferences for the resulting coupled system.." It appears that the authors have limited themselves to a normative/prescriptive perspective. This is more common with impact oriented studies such as hydro-economics and

hydro-sociology, that operationalize economic and social impact concepts respectively. To broaden it and incorporate the notion of bi-directional feedbacks, I would suggest to replace the word preference by dependence.

9) By the time one reached section 4, very little opinion has been offered on how one should go about assessing resilience of socio-hydrological systems. All that has been provided is a literature review. I therefore think that the authors should present a simple mathematical toy model and opine, based on the literature presented, on how the authors would go about assessing its resilience. This will also help better explanation of section 4.

10) The notion of how adaptation exactly increases resilience is not clear, especially because a clear definition of resilience has yet to be provided. The mathematical toy example would be a great asset here. Also, how is adaptation capacity different from absorptive capacity.

11) Section 4.3: Please discuss this in context of progress made in socio-hydrological literature. Many socio-hydrology case studies have documented that coupled systems have progress through era of development followed by preservation, e.g. pendulum swing, see Kandasamy et al. (2014, HESS), van Emmerik et al. (2014, HESS) and many more!!

12) Line 6, page 12: Enhancing adaptability under climate change is a difficult problem to tackle. Just stating that one should adapt is not enough, how one can do so using socio-hydrology is needed in this opinion piece. Again, the paper appears to be a literature review of resilience concepts borrowed from other fields (which is welcome). The authors should try to go to the next level and give us some insights on how one can learn from these ideas of resilience and plan for the Knightian uncertainty ahead (under climate change) using socio-hydrology. What is unique about socio-hydrology that can help us better prepare for the future ahead (e.g. endogenization of human agency implies less dependence on scenarios of e.g. population, land cover etc. that

may help us resolve some of the uncertainty)?

Overall, the paper appears to be a bit incoherent. It has brought ideas from other disciplines but it still has to provide an opinion based on the literature review. The paper should be made more formal, e.g. by using a mathematical toy model of SHS and all the concepts, including policy implications, should be discussed in its terms.

References: Kandasamy, J., Sounthararajah, D., Sivabalan, P., Chanan, A., Vigneswaran, S., and Sivapalan, M.: Socio-hydrologic drivers of the pendulum swing between agricultural development and environmental health: a case study from Murrumbidgee River basin, Australia, Hydrol. Earth Syst. Sci., 18, 1027–1041, doi:10.5194/hess-18-1027-2014, 2014.

van Emmerik, T. H. M., Li, Z., Sivapalan, M., Pande, S., Kandasamy, J., Savenije, H. H. G., Chanan, A., and Vigneswaran, S.: Socio-hydrologic modeling to understand and mediate the competition for water between agriculture development and environmental health: Murrumbidgee River basin, Australia, Hydrol. Earth Syst. Sci., 18, 4239-4259, doi:10.5194/hess-18-4239-2014, 2014.

Also please see the HESS special issue on Predictions under change.

---

## Author Comment (AC1) · 28 Nov 2016

We thank Dr Wesselink for her thoughtful and reflective comments, providing invaluable insights from the critical social science water research domain. Nonetheless, we think there are also a number of misunderstandings about the purpose of our paper and it positioning for readers of HESS. Consequently, in this reply we clarify our perspective on the 'problem', key contributions and approach, as well as setting out changes in response to Dr Wesselink's commentary. These changes will be incorporated in the revised version of the paper.

Dr Wesselink's comments address three aspects, which we take in turn in our reply (below). These are: (1) the concept of resilience; (2) ecosystem services, and (3) usefulness of the resilience canvas for policy making.

(1) Resilience concept.

The first comment is on the concept/ definition of resilience, one of the cornerstones of the paper. We state that 'Resilience can be understood as a set of systemic absorptive, adaptive and transformative capacities (Walker et al., 2004, 2009)'. Dr Wesselink states that we have 'redefined resilience to mean "absorptive, adaptive and transformative capacities"', and asks us to explain the relationship of these three capacities. She also asks why only the first two 'capacities' are covered in the 'resilience canvas' discussion.

Resilience is a concept with a long tradition; it has a very wide range of definitions, set out in a variety of disciplinary literatures. Yet as the reviewer notes, our paper's aim 'is not to describe this variety', nor to offer 'an extensive literature review on resilience'. Rather, we attempt to examine the potential of applying resilience in socio-hydrological contexts. To achieve this, our paper analyses the constitutive elements of both socio-hydrology (water, human and human-water) and resilience.

Given the constraints of the opinion paper format, we identify the most relevant attributes of resilience from the literatures as absorptive, adaptive and transformative capacities, without seeking to re-review the whole range of resilience characteristics (source literatures cited). In fact, resilience and adaptive capacity have very similar characteristics but opaque/ambiguous interrelations, depending on authors' viewpoints. Examples include:

- Resilience is synonymous with adaptive capacity. Tompkins and Adger (2006) for example argue that 'Adaptive capacity, which is often used to refer to the set of preconditions that enables individuals or groups to respond to climate change [. . .] is a synonym for many characteristics of resilience.' (p.5)

- Resilience is a subset of adaptive capacity (e.g. Cutter et al., 2008, Fig. 1; Gallopín, 2006, Fig. 5).
- Adaptive capacity is a subset of resilience. For example, Carpenter et al (2001) argue that 'adaptive capacity is a component of resilience' (p.766).

Faced with these different formulations, we had to choose a starting point to contextualise the socio-hydrological discussion in the later parts of the paper. To summarise and clarify, in the paper we emphasise the following propositions made already about resilience in the wider literature (source literatures cited).

- Resilience is a set of system capacities (Kuhlicke, 2013; Norris et al., 2008).

- The most salient attribute of resilience is absorptive capacity (Béné et al., 2014; Walker et al., 2004).

- Adaptive capacity is a component of resilience (Carpenter et al., 2001).

- The main difference between adaptive and transformative capacities is their magnitude of change/response. Both adaptive and transformative capacities may change the systems or stability domains, but the latter is a more extreme and sometimes 'revolutionary' form. For example, Béné et al. (2014) argue that 'Eventually, if the changes required are so large that they overwhelm the adaptive capacity of the household, community or (eco)system, transformation will have to happen' (p. 602), adaptive and transformative capacities lead to 'incremental' or 'transformational' changes in response of disturbances (p. 601). Because of their similarity, this is one of the main reasons why only 'adaptive capacity' is mentioned in the 'resilience canvas'. Another key reason to focus upon absorptive and adaptive capacities is that they are more common than the transformative capacity, and a 2D space is much more practical to convey this than a 3D space.

In sum, we reflect the line of argumentation on resilience in mainstream academic debates, rather than seeking to reconsider or re-evaluate the concept itself. Moreover, we are not the first one to identify or utilise these three characteristics: see for example Béné et al., 2014, 2015; Miller et al., 2011; Pelletier et al., 2016. Hence, we believe

we have positioned the concept of releisience appropriately and added-value for the HESS readership.

(2) Ecosystem services.

Dr Wesselink agrees that 'ecosystem services are clearly another way to look from the human', but then later states that 'ecosystem services are hardly needed in the rest of the paper', and recommends we 'remove it'. We disagree; introducing ecosystem services in the paper and into socio-hydrological resilience debates is essential, for the following reasons.

First is that ecosystem services not only provide a means to 'characterise the resilience of the human-water systems', but more importantly offer a novel perspective from which to view resilience in socio-hydrological contexts, and to stimulate new discussion on this under-researched interrelation. In this way, the theoretical contributions from ecosystem services research can we think nourish the comparatively new research field of socio-hydrological resilience, while the policy-relevant nature of ecosystem services may also prove instructive in clarifying resilience-based decision-making in socio-hydrological contexts. Moreover, this novel perspective flags the potential broader connections between resilience and ecosystem services. That is to say, the paper seeks to open up new areas of inquiry based on this novel synthesis of the two literatures as a basis for further research, rather than offering definitive answers.

In the paper, we argue that 'system identities need defining before examination is undertaken of their intrinsic resilience types' (p. 6), and 'key indicators of system state need to be established' (p. 3). For relatively conventional human-water combinations such as 'Water subsystem with anthropogenic hazards' and 'Human subsystem with hydrological hazards', the system state and its key indicators are straightforward. However, defining the state and indicators of coupled systems (i.e. socio-hydrological systems) is more problematic. The paper addresses this challenge by proposing the use of ecosystem services to describe the dynamics of socio-hydrological systems. Removing reference to ecosystem services would therefore compromise the argument set out in the paper.

Furthermore, we use ecosystem services to consider the pathways to resilience in Section 4. So we argue 'Susceptible socio-hydrological systems can be strengthened [. . .] by making hydrological ecosystem services supply more robust and sustainable under current hazard regimes' (p.9 line 19). We argue that managing the supply of 'hydrological ecosystem services' is the key to improving susceptible socio-hydrological systems, instead of managing other indicators such as hydrological biodiversity, naturalness, social security, integrity or justice. Without prior discussion of ecosystem services in advance, it would be difficult to describe what needs to be achieved in a resilient socio-hydrological system. Thus, we are clear that introducing ecosystem services and connections to socio-hydrological resilience debates is essential to communicate our key research contributions.

(3) Usefulness of the 'resilience canvas' for policy making.

Dr Wesselink comments on the usefulness of the 'resilience canvas' for policy making, and asks how to connect socio-hydrological resilience as a boundary concept with targets, how the targets can be determined, who set the targets, and how to engage policy process with resilience.

We fully agree that resilience is a 'boundary object', is 'classified as wicked or unstructured problem', and is difficult to apply in practice in public policymaking. Indeed, it was precisely to grapple with this difficulty that we wrote the paper. We believe that the main contributions we make to this significant challenge are:

- Two/three dimensional features of resilience capacities. This helps to make the concept of resilience more specific for policy making. For example, 'improvement of the absorptive capacity' is to 'resist existing hydro-hazards', while enhancing adaptive and transformative capacities are to 'cope with future uncertainties', incrementally or radically (p.8).

[Figure]

- Potential linkage between socio-hydrological resilience and ecosystem services. As explained in the previous paragraphs in this response, the concept of ecosystem services injects further policy-relevance into resilience debates. Introducing ecosystem services into socio-hydrological resilience discussion offers, we argue, a means of engaging resilience thinking with future ecosystem services-based policies.

- As Dr Wesselink notes, we foreground the role of 'people' in deciding resilience goals. Besides the two quotes highlighted by the reviewer (p.7 line 1 and p.13 line 9), we also state '... using ecosystem services to measure the state of socio-hydrological systems not only reflects the "naturalness" of the hydrological system, but also human preferences for the resulting coupled system. So, a continuing supply of ecosystem services does not necessarily mean ecosystems are pristine or close to a "natural" condition, but instead reflects the preference of the human subsystem to select for particular services' (p.8 line 2). It goes without saying that expectations of what resilience is will differ among stakeholders; this is a given of all aspects of environmental social science, and finding new ways to address how this variability can be reflected within the policy process is of paramount concern. But again, our aim here is not to discuss these normative aspects. We flag one way forward on p.12 line 4, in stating that 'polycentric water governance and public participation in more centralised forms of decision-making may play important roles in building socio-hydrological resilience'. In the revised version, we will make this point more explicit and more substantive.

Dr Wesselink finds 'the discussion in Section 4.2 too general and too obvious'. We will work on this section to make the description more specific in the revision. However, again we reiterate that our main focus in this opinion paper is not to advocate definitive answers for building or enhancing resilience, but instead to demonstrate how such studies can be placed on stronger foundations by classifying and mapping resilience-based strategies through the concept of the resilience canvas.

Dr Wesselink points out that not 'everywhere the current pathway can be described as "people and water"'. Again, we concur, though there are a couple of issues to clarify

here. We do not describe the current pathway as 'people and water'. Instead, on p.10, we introduce the three stages as 'development phases of global human-water relations' by reviewing existing research, and state that 'most current water management practice is now seeking to transition from resistant to resilient strategies' (p.11 line 28). This should not however be construed as meaning that everywhere in the world at every scale is at exactly the same stage; the picture is of course far more spatially and temporally differentiated. Furthermore, as we note the 'resilience canvas' is 'a heuristic tool to design bespoke pathways to resilience' (p.9 line 16) rather than a prescribed normative 'answer' to issues arising within a current stage.

To address the above issues in full, we will make a number of alterations to the manuscript as detailed below.

- In Section 2, we will further explain why the three resilience attributes (absorptive, adaptive and transformative capacities) are selected and what their interrelationships are.

- We will enrich and expand Section 3.1 and 3.2 to make the idea of resilience in socio-hydrological context more explicit.

- We thank the reviewer for her suggestion to remove the word 'sustainable' in p.11 line 25 to restrict the discussion within the field of resilience and to avoid confusion, and will do so in the revised manuscript.

- In the revised version of the paper, we will strengthen the reasoning behind why ecosystem services are needed in the argument, especially in Section 3.

- We will improve the discussion in Section 4.2, and make it more specific.

- We will clarify and flag the importance of water governance issues (e.g. who participate, who set the goals, and how stakeholders are engaged), especially in Section 4.

- We appreciate the suggested references, and will use them where appropriate.

References

Béné, C., Newsham, A., Davies, M., Ulrichs, M. and Godfrey-Wood, R.: Review article: Resilience, poverty and development, J. Int. Dev., 26(5), 598–623, 2014.

Béné, C., Frankenberger, T. and Nelson, S.: Design, Monitoring and Evaluation of Resilience Interventions: Conceptual and Empirical Considerations, Inst. Dev. Stud., 2015(459), 26, 2015.

Carpenter, S., Walker, B., Anderies, J. M. and Abel, N.: From metaphor to measurement: Resilience of what to what?, Ecosystems, 4, 765–781, 2001.

Cutter, S. L., Barnes, L., Berry, M., Burton, C., Evans, E., Tate, E. and Webb, J.: A place-based model for understanding community resilience to natural disasters, Glob. Environ. Chang., 18(4), 598–606, 2008.

Gallopín, G. C.: Linkages between vulnerability, resilience, and adaptive capacity, Glob. Environ. Chang., 16(3), 293–303, 2006.

Kuhlicke, C.: Resilience: A capacity and a myth: Findings from an in-depth case study in disaster management research, Nat. Hazards, 67(1), 61–76, 2013.

Miller, F., Osbahr, H., Boyd, E., Thomalla, F., Bharwani, S., Ziervogel, G., Walker, B., Birkmann, J., Leeuw, S. van der, Rockström, J., Hinkel, J., Downing, T., Folke, C. and Nelson, D.: Resilience and vulnerability: Complementary or conflicting concepts, Ecol. Soc., 15(3), 11, 2011.

Norris, F. H., Stevens, S. P., Pfefferbaum, B., Wyche, K. F. and Pfefferbaum, R. L.: Community resilience as a metaphor, theory, set of capacities, and strategy for disaster readiness, Am. J. Community Psychol., 41(1–2), 127–150, 2008.

Pelletier, B., Hickey, G. M., Bothi, K. L. and Mude, A.: Linking rural livelihood resilience and food security: an international challenge, Food Secur., 8(3), 469–476, 2016.

Tompkins, E. L. and Adger, W. N.: Does Adaptive Management of Natural Resources

[Figure]

Enhance Resilience to Climate Change?, Ecol. Soc., 9(2), 10, 2006.

Walker, B., Hollin, C. S., Carpenter, S. R., Kinzig, A., Holling, C., Carpenter, S. R. and Kinzig, A.: Resilience, adaptability and transformability in social-ecological systems, Ecol. Soc., 9(2), 2004.

---

## Author Comment (AC2) · 21 Dec 2016

We thank the anonymous reviewer for their very constructive and insightful comments, and are pleased they recognise the 'paper [makes] a valuable contribution to socio-hydrology' debates. To clarify, the main purpose of our opinion paper is to encourage debate on socio-hydrology and its interrelations with resilience, and consequently, we're pleased that the reviewer recognises there are multiple perspectives in socio-hydrology to examine, and that there's value in seeing how these perspectives might be brought into closer engagement with resilience. We take the two thoughtful reviews we have already received on the paper as indicative that this objective has been at least partially achieved.

Referring to RC2's specific comments, here we briefly clarify our opinions and arguments in relation to socio-hydrology debates. We also set out how we will address the reviewer's comments through proposed changes that will be incorporated in the revised paper.

(1) Socio-hydrology, hydro-sociology, and bi-directional feedbacks. We agree with the reviewer that the coupled human-water system can be further classified into socio-hydrological systems and hydro-social systems, each of which has different emphasises. We also fully agree that bi-directional feedbacks are the main source of resilience for human-water couplings.

However, we think extensive discussion of the differences between the two perspectives on the coupled human-water system (i.e. socio-hydrological system and hydro-sociological system) would go beyond the remit of an opinion paper. To reiterate, our aim with this paper is to encourage debate on socio-hydrology and its interrelations with resilience by conceptualising the different linkages between them. This is based on our argument that defining the system type ('coupled human-water systems') is vital to understanding their intrinsic resilience properties, and to defining indicators of their status. We proceed to argue that 'hydrological ecosystem services' can act as a status indicator of one type of bi-directional feedback. However, we emphasise that this does not mean that 'hydrological ecosystem services' offer the only means of measuring coupled human-water systems. For example, as the reviewer states, post-positivist questions such as who defines systems and who prescribes their desired state are not socio-hydrological specific, but also exist in water sub-systems with hydrological resilience, and human sub-systems with social resilience.

(2) The reviewer questions why we discuss the three types of human-water couplings in an 'isolated' way, and encourage us to 'phrase all 3 types of resilience given in section 2 in context of the larger SHS'. We see the logic of this approach to socio-hydrological thinking: indeed, this is part of the reason why this opinion paper builds its connections with the resilience concept. However, the three types of coupling encapsulate how different fields (e.g. conservation and disaster management) deal with human-water

couplings, instead of normative expectations of what people should (or should not) do.

What we propose to do:

- We will clarify and flag the post-positive and water governance issues water governance issues (e.g. who participate, who set the goals, and how stakeholders are engaged), especially in Section 4.

- We agree with the reviewer that hazards such as pollution are not always a short term exogenous perturbation, arguing instead that whether it is fast variable, frequent fast variable or slow variable depends on the scale and characteristics of the problem. We will clarify this in the revision.

- We will clarify the connections between absorptive, adaptive and transformative capacities, as well as how adaptation can be increased.

- We will redraft the sentence in Section 3.2, line 3, page 7 '... argue that the dynamics of social change should be better framed as part of socio-ecological research' to make it more specific.

- We will clarify what we mean by 'compositional approach'.

- We will replace 'preference' by 'dependence' on page 8.

- We thank the referee for recommending relevant academic papers and will cite these where appropriate in Section 4.3.

---

## Author Response (AR1)

**Authors' response to the reviewer's comments**

We thank Dr Wesselink and the anonymous reviewer for their constructive comments on the manuscript. Based on the previous replies to the reviewers, we give a point-by-point response to all comments, and indicate the changes we have made in the revised manuscript. We believe that the changes have well addressed the reviewers' comments, and consequently made significant improvements on the manuscript.

**Response to RC1 (Dr Wesselink)**

…

**Reviewer's comment:**

My first major problem is that the authors state that (p. 3) 'Resilience can be understood as a set of systemic absorptive, adaptive and transformative capacities (Walker et al., 2004, 2009)'. I do not need to go beyond reading the titles of the two references to know that Walker used the concepts 'resilience, adaptability and transformability'. So instead of resilience as one of three system characteristics (as Walker does), the authors use resilience as overarching concept that includes the other two characteristics, and 'absorptive capacity equates to the original concept of resilience' (p.4). They thereby re-define a very well-known concept. This is never a good idea, and certainly not in an article whose main theme is 'conceptual clarity'. I therefore wonder why the authors redefined resilience to mean 'absorptive, adaptive and transformative capacities', what their relationships are, and how they compose together resilience? Re-definitions lead to confusion rather than clarity. Since I am not an expert on resilience studies, I cannot estimate what this re-definition does in terms of changing what is studied and on what terms. I hope a reviewer with more knowledge in the SES-resilience domain can shed light on this.

*Authors' response:*

*As explained in the preliminary response, we did not 'redefine' the resilience concept as 'absorptive, adaptive and transformative capacities', but followed a popular definition of resilience – 'In popular terms, resilience is having the capacity to persist in the face of change, to continue to develop with ever changing environments'* (Folke, 2016, p.2)*, and 'resilience as the result of absorptive, adaptive and transformative capacities'* (Béné et al., 2014, p.601)*. In fact, we reflect the line of argumentation on resilience in mainstream academic debates, rather than seeking to reconsider or re-evaluate the concept itself.*

*To address this comment, in Section 2 of the revised manuscript, we further explained the resilience concept and the interrelationships of the three capacities.*

**Reviewer's comment:**

I think the treatment of the meaning of resilience in the one way views on human water systems (sections 3.1 and 3.2) is rather brief. I would especially like to see the application pf the three elements of resilience that the authors defined, i.e. absorptive, adaptive and transformative capacities. In fact, in 31 the authors themselves the 'old' understanding of resilience when they discuss the water subsystem with anthropogenic hazards when they state that (p.5) 'the implicit goal of maintaining subsystem equilibrium or restoring it to a desired historical state' leads to 'resilience management' –

surely in the authors' definitions this should read 'managing absorptive capacity'? In section 3.2 the same applies, as again in section 4.3.

*Authors' response:*

*We have enriched and expanded Section 3.1 and 3.2, and added one paragraph at the end of each Section to clarify how the three capacities can be applied and improved.*

**Reviewer's comment:**

Related to the above, why does transformative capacity not appear in the 'resilience canvas'? Is this maybe a possibility for coping that is beyond or against the authors' own norms and aspirations? If this is the case, it needs to be made explicit; however, the internal logic of the theoretical argument loses much strength: first resilience is three-part, then only two parts are used to assess resilience (absorptive and adaptive capacity).

*Authors' response:*

*To follow the reviewer's comments, in Section 4.1, we explained why only the first two capacities (i.e. absorptive and adaptive) are compared in the presentation of the 'resilience canvas', and explicitly stated that transformative capacity is beyond the aspirations of this paper. The two main reasons are as follows.*

*(1) There is still an ongoing debate on what exact systematic attributes are needed to support a radical transformation to an entirely new stage (Robinson and Carson, 2015; Wilson et al., 2013).*

*(2) We would like to keep the analysis of resilience capacities in a visually simple way as a 2-dimensional space instead of a 'resilience cube', and thus selected the first two better studied capacities for demonstration purposes.*

*Therefore, transformative capacity is not the main emphasis in the 'resilience canvas' section, but has great potentials to be explored in the future.*

**Reviewer's comment:**

My second major problem is the substitution of 'ecosystem services' for 'resilience' when assessing two-way interactions in human-water systems. In the justification that is presented, ecosystem services are clearly another way to look from the human to the natural system (p.8): 'a continuing supply of ecosystem services does not necessarily mean ecosystems are pristine or close to a 'natural' condition, but instead reflects the preference of the human subsystem to select for particular services'. It turns out that ecosystem services are hardly needed in the rest of the paper, so I suggest to remove it. However, this does leave one of the goals of the paper unfulfilled: to characterise the resilience of the human-water systems. In my view, a major re-think is needed here, or a reduction of the ambitions of the paper. The latter would be perfectly acceptable, since the conceptual content is considerable, as expressed especially in the figures.

*Authors' response:*

*In Section 3.3, we added some sentences in the second paragraph to explain the necessity of having 'ecosystem services' in the discussion which was explained in the previous response:*

*First is that ecosystem services not only provide a means to 'characterise the resilience of the human-water systems', but more importantly offer a novel perspective from which to view resilience in socio-*

*hydrological contexts, and to stimulate new discussion on this under-researched interrelation. In this way, the theoretical contributions from ecosystem services research can we think nourish the comparatively new research field of socio-hydrological resilience, while the policy-relevant nature of ecosystem services may also prove instructive in clarifying resilience-based decision-making in socio-hydrological contexts. Moreover, this novel perspective flags the potential broader connections between resilience and ecosystem services. That is to say, the paper seeks to open up new areas of inquiry based on this novel synthesis of the two literatures as a basis for further research, rather than offering definitive answers.*

*In the paper, we argue that 'system identities need defining before examination is undertaken of their intrinsic resilience types', and 'key indicators of system state need to be established'. For relatively conventional human-water combinations such as 'Water subsystem with anthropogenic hazards' and 'Human subsystem with hydrological hazards', the system state and its key indicators are straightforward. However, defining the state and indicators of coupled systems (i.e. socio-hydrological systems) is more problematic. The paper addresses this challenge by proposing the use of ecosystem services to describe the dynamics of socio-hydrological systems. Removing reference to ecosystem services would therefore compromise the argument set out in the paper.*

*Furthermore, we use ecosystem services to consider the pathways to resilience in Section 4. So we argue 'Susceptible socio-hydrological systems can be strengthened [...] by making hydrological ecosystem services supply more robust and sustainable under current hazard regimes'. We argue that managing the supply of 'hydrological ecosystem services' is the key to improving susceptible socio-hydrological systems, instead of managing other indicators such as hydrological biodiversity, naturalness, social security, integrity or justice. Without prior discussion of ecosystem services in advance, it would be difficult to describe what needs to be achieved in a resilient socio-hydrological system. Thus, we are clear that introducing ecosystem services and connections to socio-hydrological resilience debates is essential to communicate our key research contributions.*

**Reviewer's comment:**

My final comments relate to the usefulness of the paper/the resilience canvas for policy making. One element of this discussion relates to the (perceived by the authors) need to specify what resilience means. This is commonly known as an activity of framing or structuring, with resilience classified as wicked or unstructured problem (Hisschemoller & Hoppe 1995; Hisschemoller et al 2001; Hoppe 2008). The authors seem to be aware of this (p.13): 'Resilience is not only a descriptive notion, and usually has normative (goal-setting) objectives'; also in section 3.2 'Resilience from the perspective of managing human subsystems thus emphasises particular societal norms and goals or normative aspirations in relation to hydrological hazards'. At the same time, they do not question the notion of 'desired state': desired by whom? In the conduct of research, it will be the researchers who decide what is desired (often based on ecological arguments). In other words: targets for the water system would be just as dependent on societal norms and aspirations!

*Authors' response:*

*We agree with the reviewer's comments that the targets could be 'dependent on societal norms and aspirations'. However, the 'desired states' also depend on what socio-hydrological type they are in. To address this comment, we compared how 'desired systematic states' are defined in all three framings throughout Section 3, especially Section 3.3. We argued that 'desired states of the water sub-system are usually high naturalness or historical conditions measured by biotic and abiotic indicators, while desired states of the human sub-system are more normative societal expectations set by relevant social groups'. In addition, 'using ecosystem services to measure the state of socio-*

*hydrological systems not only reflects the "naturalness" of the hydrological system, but also human preferences for the resulting coupled system'.*

**Reviewer's comment:**

Problem structuring is done when resilience is used within research, but even more so when it is stated as a policy objective, since it can be expected that stakeholders' values and interests diverge more than researchers'. I therefore question the notion that researchers should prescribe (i.e. pre-structure) how policy processes should engage with boundary or nirvana concepts like resilience – this should be done in the policy process. In fact, the literature on boundary objects (Molle 2008; Walker & Shove 2007) suggests that interpretative flexibility of goals is essential in policy making.

*Authors' response:*

*We agree with the reviewer that interpretative flexibility of goals is essential for boundary objects in policy making. The classification of three framings is actually supporting this flexibility instead of avoiding it, by clarifying that different people may use one of the three framings in the socio-hydrological context according to their needs. In addition, we highlighted that the expected states for human sub-systems and socio-hydrological systems are set by relevant social groups, and encouraged public participation in the process of determining the goal.*

*In the Introduction Section of the revised manuscript, we elaborated the aim of the paper, which is 'to propose a conceptual framework for assessing resilience in socio-hydrological contexts, and by which we provide opinions for understanding and managing socio-hydrological resilience. Instead of offering a single prescriptive solution, this framework supports pluralist perspectives and encourages debate on socio-hydrology and its interrelations with resilience.'*

**Reviewer's comment:**

While I am fascinated by the way in which the authors depict pathways on the resilience canvas, I find the discussion in section 4.2 too general and too obvious. Neither the resilience canvas (or its derivation) nor the concept of socio-hydrology is necessary to make these general statements. Section 4.3 is again interesting [note: this is the only place ecosystems services make a brief appearance, not just as needing to be resilient but also sustainable (p.11 line 25): mentioning this term opens another can of worms at least as large as the resilience confusion!]. However, I do disagree that everywhere the current pathway can be described as 'people & water'. For example, the EA for England & Wales [NB: not UK!] approach to flood risk management in fact increases individual people's vulnerability by transferring responsibility to individuals rather than the EA.

*Authors' response:*

*We have enriched and expanded the discussions in Section 4.2, and provided a new example (comparison of two catchments as well as a new Figure 4) to make it more specific.*

*We have removed the word 'sustainable' in the 'People and Water' subsection to avoid confusion.*

*As mentioned in the previous reply, we do not describe the current pathway as 'people and water'. Instead, we introduce the three stages as 'development phases of global human-water relations' by reviewing existing research, and state that 'most current water management practice is now seeking to transition from resistant to resilient strategies'. This should not however be construed as meaning that everywhere in the world at every scale is at exactly the same stage; the picture is of course far more spatially and temporally differentiated.*

*Thanks for pointing out the typo 'the UK's Environment Agency', and we have changed it to 'Environment Agency for England & Wales'.*

*It is debatable whether the EA has transferred ALL responsibility to individuals, or sharing responsibility with individuals. However, it could be understood as an example of increasing individual's resilience by public participation, opening the risk information and using information and communication technologies.*

**Response to RC2**

**Reviewer's comment:**

1) The authors appear to treat hydro-sociology and socio-hydrology as if these two fields are similar in their demand to understand resilience (or possible instability) in coupled human water systems. The concept proposed by the authors is more formal and requires a post-positivist approach, which seeks regularities and generalizable relationships between human and their water system contingent on disciplines involved. It is not clear where hydro-sociology stands in this context, if it is anti-positivist (i.e. not seeks generalizable relationships) then it will be difficult to interpret outcomes of nonlinear system dynamics such as instability and resilience. Socio-hydrology offers not only a quantitative framework but such a framework that allows for bi-directional feedbacks. I wonder if hydro-sociology is more oriented towards impact assessment and implementation of social objectives or outcomes – and bi-directional feedbacks may be key to understanding resilience of SHS. The authors may want to clarify the positioning and capacity of hydro-sociology in being able to implement the formal concept of resilience that is being presented here.

*Authors' response:*

*As discussed in the previous response, we agree with the reviewer that the coupled human-water system can be further classified into socio-hydrological systems and hydro-social systems, each of which has different emphases. We also fully agree that bi-directional feedbacks are the main source of resilience for human-water couplings. However, we think extensive discussion of the differences between the two perspectives on the coupled human-water system (i.e. socio-hydrological system and hydro-sociological system) would go beyond the remit of an opinion paper.*

*In the revision, we emphasised in Section 3 that this paper follows the Sivapalan's interpretation of socio-hydrology which has the focus on the co-evolution and feedbacks of coupled human-water systems to avoid any misunderstandings* (Sivapalan et al., 2012)*.*

**Reviewer's comment:**

2) Why should we consider the resilience and stability of the hydro and socio subsystems of a SHS if it doesn't affect the bi-directional feedbacks. It runs very contrary to the philosophy of socio-hydrology which emphasizes that endogenization of human agency as its key consideration. Why should we even be bothered to study instabilities isolated to certain sub-systems when it does not spread to the larger system through its coupled dynamics? Why should we study instabilities in hydrological systems when they are of no concern to humans – current or future! If such instabilities are of concern to humans then we are talking about resilience of the entire SHS, not isolated subsystems. I therefore encourage the authors to phrase all 3 types of resilience given in section 2 in context of the larger SHS and to not isolate them.

*Authors' response:*

*As explained in the previous response, we see the logic of this approach to socio-hydrological thinking: indeed, this is part of the reason why this opinion paper builds its connections with the resilience concept.*

*In the revised manuscript, we further explained why three types of resilience framings are classified in the introduction paragraph of Section 3. We argued that 'socio-hydrological resilience should refer to resilience of socio-hydrological systems which is one specific type of resilience in socio-hydrological contexts. The former two types focus on intrinsic hazard-subsystem relations, while the latter covers these subsystem relations and broader and more iterative interplay between them.' In addition, 'the three types of coupling encapsulate how different fields (e.g. conservation, disaster management and water resources management) deal with human-water couplings, instead of normative expectations of what people should (or should not) do'.*

**Reviewer's comment:**

3) Same as in point 2, the notion that hazard such as pollution (section 3.1) is a short term exogenous perturbation is incorrect in my opinion. It is caused by humans, and human respond to that through community sensitivity and change their norms. Thus hazard is not an exogenous perturbation but an endogenous one as a result of human agency. It is also not a short term perturbation since its effect through community sensitivity can be over decades, even centuries. See for van Emmerik et al. (2014, HESS) and Kandasamy et al. (2014, HESS). I therefore think isolating the concept of resilience to hydro and socio sub-systems is contrary to socio-hydrology and misguided.

*Authors' response:*

*For the reasons given in the above response to Comment 2, whether pollution is an exogenous or endogenous perturbation depends on how resilience is framed in the socio-hydrological context – pollution is more likely to be an exogenous perturbation for water sub-systems, but an endogenous one for socio-hydrological systems. In addition, we agreed that calling pollution as a 'short term perturbation' is misleading, and changed it into 'occasional, recurrent and continuous perturbations' in the second paragraph of Section 2.*

**Reviewer's comment:**

4) The discussion of absorptive, adaptive and transformative capacity is not clear. I am surprised that the authors decided to discuss such formal concepts in a qualitative manner. To ensure tractability of concepts, I would encourage authors to present a mathematical toy problem and express these concepts more formally.

*Authors' response:*

*In the revised manuscript, we improved the discussion of the three capacities and their relationships.*

*- In Section 2, we further explained where the three capacities come from. We argued in the first paragraph of Section 2 that, in a popular term, 'resilience is "the capacity to persist in the face of change, to continue to develop with ever changing environments" (Folke, 2016 p.2). Thus, this notion is understood as a set of systemic absorptive, adaptive and transformative capacities, which provide a nuanced conceptualisation in three dimensions – persistence for now, response for future contingencies in incremental or radical ways (Béné et al., 2014; Miller et al., 2011).'*

*- In Section 3, we provided examples of the three capacities for each type of resilience framing in the last paragraphs of each subsection (Section 3.1, 3.2 and 3.3).*

*- In the second paragraph of Section 4.1, we explicitly explained why the first two capacities (i.e. absorptive and adaptive capacities) were selected to construct the 'resilience canvas'.*

*The discussions of the three capacities reflect the line of argumentation on resilience in mainstream academic debates, and follow some of the most influential papers in the field (see e.g. Béné et al., 2014; Folke, 2016; Walker et al., 2004), which are also in a qualitative manner. We agreed with the reviewer that some mathematical toy problems will be useful to express these concepts more formally. However, providing quantitative expressions of resilience capacities that have seldom appeared in the existing papers might be beyond the scope of this opinion paper. Nevertheless, in the concluding remarks section, we argued that new quantification approaches and mathematical tools are required as future work.*

**Reviewer's comment:**

5) Page 6, Line 27, "..resilience is to use a more theoretically pluralist perspective..": totally agree, and socio-hydrology welcomes such a perspective.

*Authors' response:*

*Thanks for agreeing with this as one of the main purposes of the paper – to encourage pluralist perspectives and debates on the inter-connections between socio-hydrology and resilience.*

**Reviewer's comment:**

6) Section 3.2, line 3, page 7 ".. argue that the dynamics of social change should be better framed as part of socio-ecological research.": if you extrapolate it to sociohydrological systems, then this is more about coupling and bi-directional feedbacks between human and their water system, contrary to the idea of the section that resilience should be treated in (social sub-system) isolation with hydrological hazards as (exogenous) boundary conditions.

*Authors' response:*

*We agreed with this comment and have reorganised our summary of Cote and Nightingale's work (2012), changing from '.. argue that the dynamics of social change should be better framed as part of socio-ecological research' to 'argue that there is still far less attention to normative and epistemological questions'.*

**Reviewer's comment:**

7) When I come to section 3, it is not clear where the authors stand in terms of how resilience should be studied (whether in isolation or not). They evoke arguments against the compositional approach but their resilience framing appears to be compositional in nature! The authors should clarify this, I hope in favor of abandoning studying resilience of sub-systems.

*Authors' response:*

*We agreed with the reviewer that the description of the compositional approach in the original version was not clear, and revised Section 3.3. In the revised manuscript, we clarified that the two approaches are to measure the 'state of systems' instead of 'resilience of systems'. It is because defining*

*systematic states helps to explain and clarify the identity of resilience, and answer the classic 'resilience of what' question. Therefore, the differences between the two approaches are now clearer.*

*- A compositional approach is to assess the state of coupled systems by summing up separately assessed 'system components'.*

*- An ecosystem services approach is to directly assess hydrological ecosystem services as one of key human-water interactions. In this method, the 'coupled systems' are not broken down to components to be assessed separately. Although the general hydrological ecosystem services may be comprised of a couple of sub-services (e.g. provisioning, regulating and cultural), the assessment of each sub-service is still a direct evaluation of a certain facet of human-water interactions.*

**Reviewer's comment:**

8) Lines 3-4, page 8: "..human preferences for the resulting coupled system.." It appears that the authors have limited themselves to a normative/prescriptive perspective. This is more common with impact oriented studies such as hydro-economics and hydro-sociology, that operationalize economic and social impact concepts respectively. To broaden it and incorporate the notion of bi-directional feedbacks, I would suggest to replace the word preference by dependence.

*Authors' response:*

*We accepted this suggestion and replaced the word 'preference' by 'dependence' in Section 3.3.*

**Reviewer's comment:**

9) By the time one reached section 4, very little opinion has been offered on how one should go about assessing resilience of socio-hydrological systems. All that has been provided is a literature review. I therefore think that the authors should present a simple mathematical toy model and opine, based on the literature presented, on how the authors would go about assessing its resilience. This will also help better explanation of section 4.

*Authors' response:*

*For the reasons mentioned in the response to Comment 4, we thought mathematical toy models might be beyond the scope of this opinion paper and quantification assessment methods should be developed in the future to transform the 'resilience canvas' from a conceptual tool to a quantifiable one. However, we have enriched the contents of Section 4 and made the opinions more explicit. In summary, the main opinions in Section 4 include:*

*- Pathways to resilience can be designed with help of the newly proposed 'resilience canvas'.*

*- Systems with similar overall resilience evaluation may differ in the composition of their composition of resilience capacities. It implies that bespoke strategies should be developed for each system to make the pathways effective.*

*- The constitutive capacities of resilience do not usually grow equally while the overall resilience is increasing. It means that the pathways are not always in straight lines.*

*- In general, we need to shift from resistant to resilient water management strategies.*

**Reviewer's comment:**

10) The notion of how adaptation exactly increases resilience is not clear, especially because a clear definition of resilience has yet to be provided. The mathematical toy example would be a great asset here. Also, how is adaptation capacity different from absorptive capacity.

*Authors' response:*

*To answer how adaptation increases resilience, as well as how adaptive capacity differs from absorptive capacity, we could refer to the popular definition of resilience introduced in Section 2 - 'resilience is "the capacity to persist in the face of change, to continue to develop with ever changing environments" (Folke, 2016 p.2). Thus, this notion is understood as a set of systemic absorptive, adaptive and transformative capacities, which provide a nuanced conceptualisation in three dimensions – persistence for now, response for future contingencies in incremental or radical ways (Béné et al., 2014; Miller et al., 2011).'*

*We have improved the discussion of the relationship between absorptive (resilience) and adaptive capacities, please refer to the response to Comment 4.*

*For mathematical toy examples, please also refer to the response to Comment 4.*

**Reviewer's comment:**

11) Section 4.3: Please discuss this in context of progress made in socio-hydrological literature. Many socio-hydrology case studies have documented that coupled systems have progress through era of development followed by preservation, e.g. pendulum swing, see Kandasamy et al. (2014, HESS), van Emmerik et al. (2014, HESS) and many more!!

*Authors' response:*

*Thank you for suggesting the case studies for the 'resilience canvas'. Although these studies are about the development of 'socio-hydrological systems', but Section 4.3 is about the development of 'socio-hydrological resilience', significant overlaps between the suggested papers and this one can be found. We used the suggested cases to enrich the contents of Section 4.3. Please refer to the marked-up manuscript for details (especially the second and last paragraph of Section 4.3).*

**Reviewer's comment:**

12) Line 6,page 12: Enhancing adaptability under climate change is a difficult problem to tackle. Just stating that one should adapt is not enough, how one can do so using socio-hydrology is needed in this opinion piece. Again, the paper appears to be a literature review of resilience concepts borrowed from other fields (which is welcome). The authors should try to go to the next level and give us some insights on how one can learn from these ideas of resilience and plan for the Knightian uncertainty ahead (under climate change) using socio-hydrology. What is unique about socio-hydrology that can help us better prepare for the future ahead (e.g. endogenization of human agency implies less dependence on scenarios of e.g. population, land cover etc. that may help us resolve some of the uncertainty)? Overall, the paper appears to be a bit incoherent. It has brought ideas from other disciplines but it still has to provide an opinion based on the literature review. The paper should be made more formal, e.g. by using a mathematical toy model of SHS and all the concepts, including policy implications, should be discussed in its terms.

*Authors' response:*

*Instead of discussing how 'adaptive capacity' can be practically enhanced under climate change, this opinion paper chose to focus on urgent but easily neglected conceptual questions about 'adaptive capacity' in at least four ways.*

*(1) What is the relationship between adaptive capacity and resilience?*

*We examined the definition of resilience and regarded adaptive capacity as one of the three resilience capacities to cope with future contingencies by incremental improvement and development (Section 2).*

*(2) Where do adaptive capacities come from?*

*We identified three framings of resilience and human-water couplings, and argued that adaptive capacity may come from water sub-system, human sub-system or human-water interactions. For each resilience source, there are specific strategies to enhance adaptive capacity (Section 3, last paragraphs of Section 3.1, 3.2 and 3.3).*

*(3) What is the role of adaptive capacity in water management under change?*

*We advocated a switch of general water management strategies from resistant to resilient building by improving adaptivity. This provides an important conceptual basis to consider adaptive strategies in the resilience context (Section 4).*

*(4) What are the emerging or promising fields that need specially attentions and potentially support the new adaptivity-oriented water management strategy?*

*Three fields have been identified (Section 4.3):*

*- a reemphasis on the ecosystem integrity, which is aligned with the 'pendulum swing' phenomenon (Kandasamy et al., 2014).*

*- polycentric water governance (Buytaert et al., 2014, 2016)*

*- information and communication technologies (Karpouzoglou et al., 2016)*

*Again, as discussed in the above response, in the revised manuscript, we tried to make the opinions more explicit, the structure more coherent, the argument clearer, and discussed future works that can take this conceptual framework forward. We believed these efforts have contributed to the achievement of the purpose, which is to encourage debates 
[revised manuscript text omitted]